# PMEL is involved in snake colour pattern transition from blotches to stripes

Athanasia C. Tzika [1] ✉, Asier Ullate-Agote [1,3], Pierre-Yves Helleboid[1] & Maya Kummrow[2]

Corn snakes are emerging models for animal colouration studies. Here, we focus on the Terrazzo morph, whose skin pattern is characterized by stripes rather than blotches. Using genome mapping, we discover a disruptive mutation in the coding region of the *Premelanosome protein* (*PMEL*) gene. Our transcriptomic analyses reveal that *PMEL* expression is significantly down-regulated in Terrazzo embryonic tissues. We produce corn snake *PMEL* knockouts, which present a comparable colouration phenotype to Terrazzo and the subcellular structure of their melanosomes and xanthosomes is also similarly impacted. Our single-cell expression analyses of wild-type embryonic dorsal skin demonstrate that all chromatophore progenitors express *PMEL* at varying levels. Finally, we show that in wild-type embryos *PMEL*-expressing cells are initially uniformly spread before forming aggregates and eventually blotches, as seen in the adults. In Terrazzo embryos, the aggregates fail to form. Our results provide insights into the mechanisms governing colouration patterning in reptiles.

Blotches, spots, labyrinths, stripes, and any combination of these are among the patterns encountered in animal skin colouration. In-depth work has been done on classical model species to identify the key players involved in the establishment of these spectacular phenotypes[1] and to understand how these elements interact[2,3]. In zebrafish, a species whose skin colouration has been extensively studied, it is the interactions of three types of cells in the skin—the melanophores, the xanthophores, and the iridophores—that pro-duce the yellow and black stripes of the adult[4]. These cells originate from the neural crest and form the larva skin pattern during embryogenesis, and then the adult pattern during metamorphosis. The recent development of single-cell transcriptomics makes it possible to investigate the gene expression profile of each chro-matophore type at different developmental stages, thus furthering our understanding of how they differentiate and mature[5,6]. By elu-cidating the mechanisms that bring about the wild-type zebrafish colouration, it is also possible to understand the phenotypic diversity of other teleosts[7,8], but it remains to be seen if the extensive findings in this vertebrate lineage can be extrapolated to the other vertebrate lineages[9].

Reptiles, comprising more than 12,000 species[10], exhibit equally spectacular colouration phenotypes as teleost fish but the inherent difficulties in maintaining and breeding them in captivity, mainly due to their larger size and the long generation time, has restricted the number of studies on these animals. Yet a handful of squamate species (snakes and lizards) are currently promoted as new models. In recent years, several colour morphs have been characterised for the corn snakes (*Pantherophis guttatus*)[11,12], the leopard geckos (*Eublepharis macularius*)[13,14], and the ball pythons (*Python regius*)[15,16]. These animals have been kept in captivity for decades by private breeders and numerous spontaneously-occurring mutations have appeared, affect-ing both their colouration and their colouration pattern. The sys-tematic genetic characterisation of these colour morphs, coupled with developmental, transcriptomic, and functional analyses, will help us discover the mechanisms that establish the patterning of squamate chromatophores.

[1]Laboratory of Artificial & Natural Evolution (LANE), Department of Genetics & Evolution, University of Geneva, Geneva, Switzerland. [2]Tierspital, University of Zurich, Zurich, Switzerland. [3]Present address: Biomedical Engineering Program, Center for Applied Medical Research (CIMA), Universidad de Navarra, Instituto de Investigación Sanitaria de Navarra (IdiSNA), Pamplona, Spain. ✉e-mail: athanasia.tzika@unige.ch

In this work, we focus on Terrazzo, a corn snake colour morph that presents stripes instead of blotches (Fig. 1). We genetically characterize this morph using a mapping-by-sequencing approach. We shortlist 35 candidate genes in a 1.9 Mb genomic interval. Most of these genes carry fixed-point mutations that minimally affect their coding sequence and functional domains. Only a multi-nucleotide polymorphism in the *Premelanosome protein* (*PMEL*) results in the disruption of the protein sequence. We also characterize the different *PMEL* isoforms expressed in Terrazzo individuals and analyse the *PMEL* expression profile both with bulk RNA sequencing and real-time quantitative PCR in the developing skin and other embryonic tissues of Terrazzo and wild-type individuals. To confirm that *PMEL* is involved in the proper pattern establishment in corn snakes, we produce *PMEL* corn snake knockouts (KO), that display striped phenotypes, like the Terrazzo individuals. Electron microscopy imaging of wild-type, Terrazzo and *PMEL* knockouts reveal modifications in the subcellular structure of melanophores and xanthophores. Furthermore, we perform single-cell transcriptomic analyses to investigate the *PMEL* expression profile in chromatophores, and we show that it is expressed by all chromatophore progenitors at varying levels. Whole-mount in situ hybridisations on embryos of relevant developmental stages using a *PMEL* probe document how the blotches of the wild-type animals are formed. It is not possible to detect any *PMEL* expression on the Terrazzo embryos, either because its expression level is too low, or because the number of cells expressing it is reduced. Additional whole-mount hybridisations with different chromatophore progenitors' markers, identified by our single-cell experiment, suggest that the overall number of chromatophores is reduced in Terrazzo individuals. Thus, a minimum number of

chromatophores is necessary for the formation of blotches in the wild-type. These results will serve as a starting point to elucidate the developmental mechanisms involved in reptilian colouration and have implications in the fields of evolutionary developmental biology, herpetology and ecology.

## Results

### Description of the Terrazzo corn snake morph

Wild-type corn snakes are characterised by red dorsal and lateral blotches, delineated by a black line, on an orange background and the black pupil of their eyes is surrounded by a red iris (Fig. 1a). The Terrazzo corn snake morph originates from a cross of two 'rosy rat' corn snakes. These 'rosy rat' corn snakes are natives of the Key islands in Florida and they are hypomelanistic, with reduced black pigmentation[17]. The Terrazzo hatchlings have two red dorsal stripes that run down the entire length of their body on an orange background (Supplementary Fig. 1). Two red lateral stripes are visible at the anterior part of their body, but these gradually fade towards the posterior part. The longitudinal stripes are delineated by very thin black lines, whose thickness varies along the length of the body and from one individual to the other; occasionally, they can be absent. Over the years and as the animal grows, the dorsal and lateral stripes are maintained at the anterior part of the body, and the posterior part takes up an overall speckled grey/orange colouration without a distinct pattern (Fig. 1b). The iris of the Terrazzo animals is also speckled, instead of red. Finally, the black and white checkers on the ventral scales of the wild-type are replaced by a uniform white colouration at the anterior part of the body of the Terrazzo animals that progressively turns red posteriorly (Supplementary Fig. 1). Note that

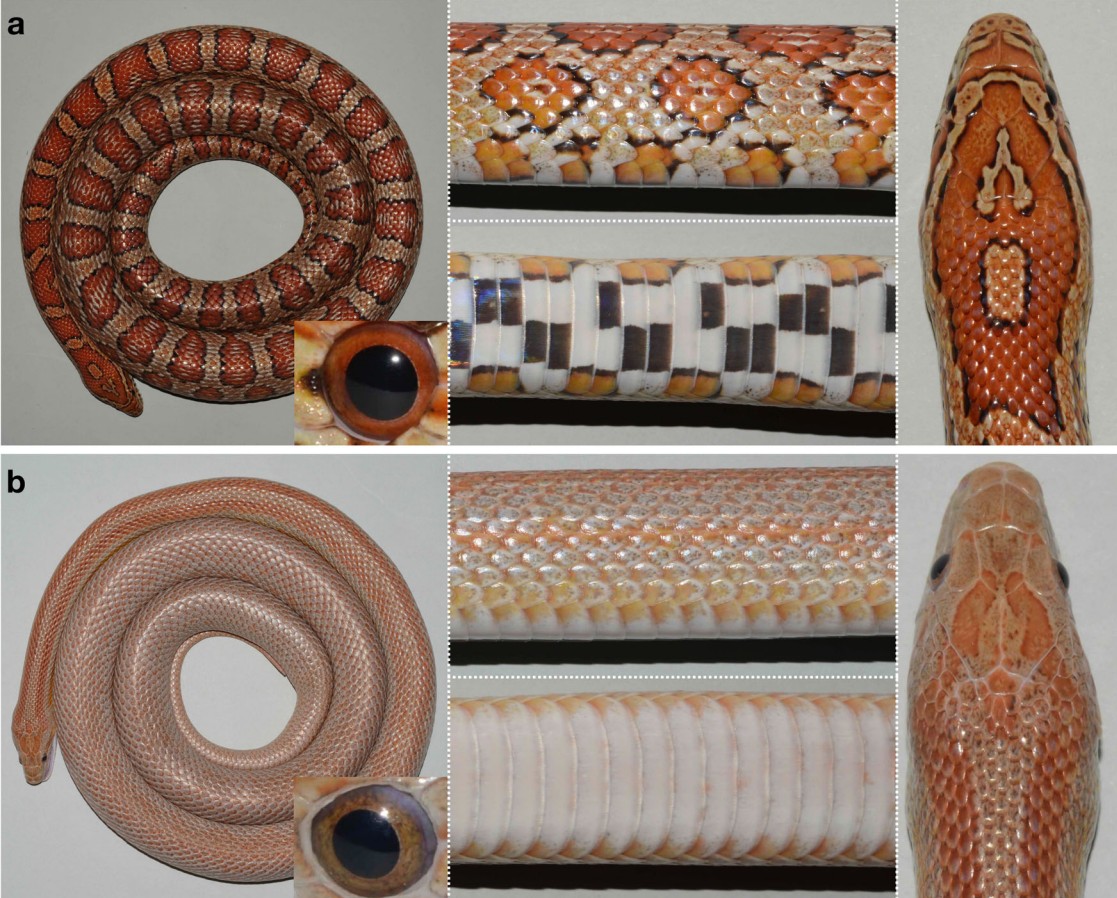

**Fig. 1 | Skin colour pattern of wild-type and Terrazzo corn snakes.** Dorsal overview (left), close-ups of lateral (top centre) and ventral views (bottom centre), and head dorsal view (right) of a wild-type (**a**) and a Terrazzo (**b**) individual. Close-ups of the eyes are provided in the insets.

red pigmentation can also be seen ventrally at the posterior body of the wild-type animals.

## Mapping of the Terrazzo variant on *PMEL*

Crosses in our facility confirmed that a single-locus recessive mutation is responsible for the Terrazzo morph[18]. We performed mapping-by-sequencing analyses to retrieve the causative variant, as previously described[19]. In short, we crossed a homozygous Terrazzo male (*tz/tz* genotype) with two heterozygous females (*tz/+*) to obtain homozygous and heterozygous offspring for the causative allele. We sequenced four genomic DNA libraries from: the homozygous male, the two heterozygous females, a pool of 26 homozygous offspring and a pool of 19 heterozygous offspring (Supplementary Table 1; NCBI accession number PRJNA1073516). We aligned each library separately to our corn snake genome assembly[19] (GCF001185365.1) and searched for single- and multi-nucleotide polymorphisms co-segregating with the Terrazzo genotype in non-repetitive elements[19]. We could identify an interval of 12.05 Mb on Super-scaffold 85 (NW_023010713; from 2.86 to 14.91 Mb; Fig. 2a, b), where there are 42,973 co-segregating SNPs/MNPs with a density of 3568 variants/Mb. We found regions with a high proportion of co-segregating variants in four additional Super-scaffolds, but not as high as in Super-scaffold 85 (Supplementary Fig. 2). We genotyped 91 individuals and identified 13 recombinants

that allowed us to reduce the interval to 1.9 Mb (from 13 to 14.9 Mb; Supplementary Fig. 3a, b). The corresponding interval on the latest chromosome-length corn snake assembly from Columbia University[20] (CU assembly; GCF_029531705.1) is 2.12 Mb long (NW_026844023.1; from 82.93 to 84.85 Mb; Supplementary Fig. 3c).

Based on the NCBI annotation of both assemblies, there are 35 protein-coding genes in the reduced interval (Supplementary Table 2). We identified indels, single- and multi-nucleotide polymorphisms that co-segregate with the Terrazzo genotype in 27 of these genes, most of which result in synonymous substitutions. There are non-synonymous amino acid modifications in the coding sequence of 14 genes, but these do not affect the domain predictions by InterProScan[21]. On the other hand, a multi-nucleotide polymorphism, from ACG to TCC (Super-scaffold 85; position 13,272,041) falls at the end of exon 8 (one nucleotide – A to T) and on the donor site of intron 8 of the *Premelanosome Protein* (*PMEL*) gene (two nucleotides – CG to CC) (Fig. 2c).

PMEL is a glycoprotein produced in melanophores and the retinal pigmented epithelium. The PMEL protein goes through substantial post-translational modifications, mainly O-glycosylation and cleaving, during and after transfer in the early-stage melanosomes. There, the PMEL fragments assemble into striated amyloid fibrils providing a scaffold for the deposition of melanin[22]. The corn snake PMEL protein is 713 amino acids (aa) long and comprises: an initial signalling peptide

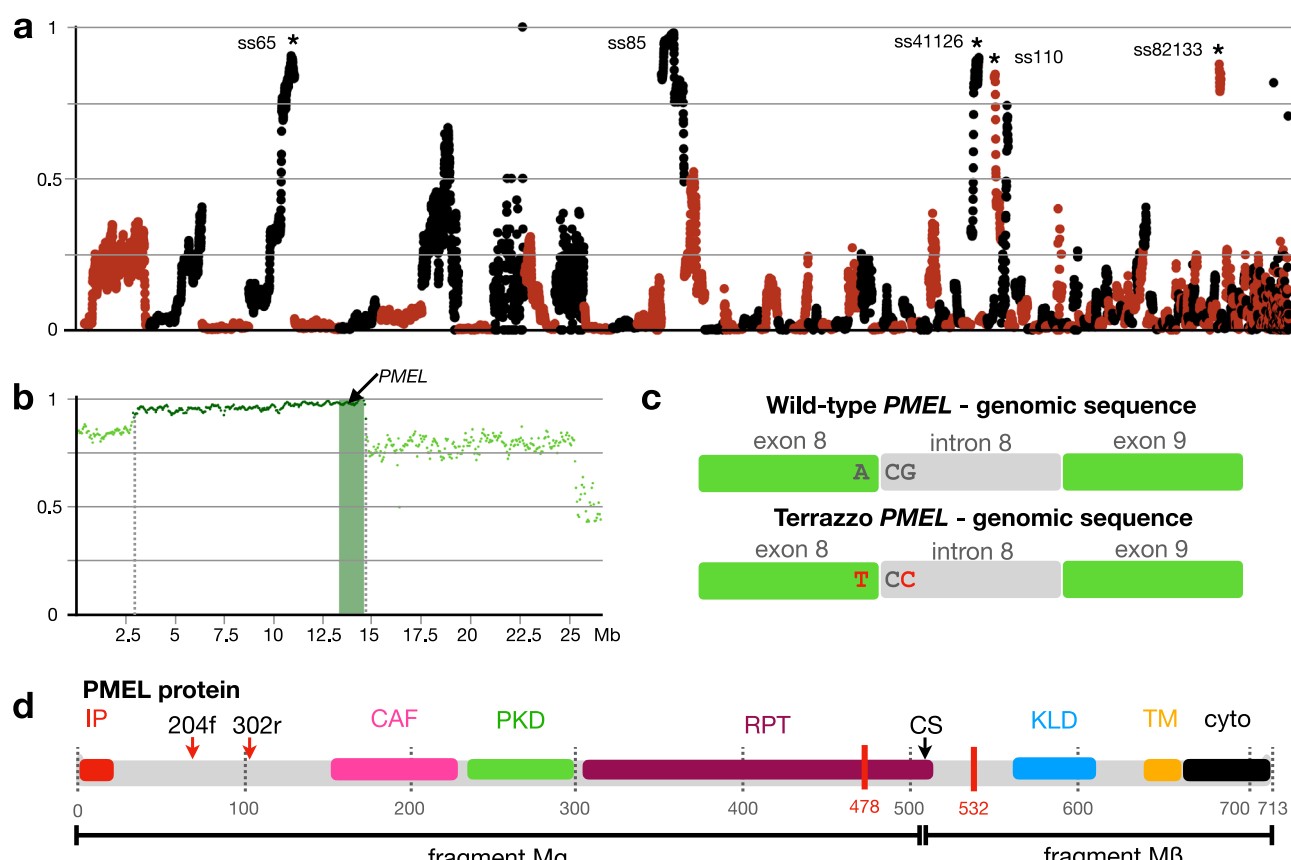

**Fig. 2 | Mapping of the Terrazzo variant. a** Proportion (y-axis) of quality-filtered biallelic SNP/MNP co-segregating with the Terrazzo locus in the four genomic libraries compared to informative quality-filtered parental variants. Proportions are calculated for scaffolds >1 Mb, with a 1-Mb sliding window and a step of 100 Kb. Scaffolds (alternatively coloured black and red) are ordered from longest to shortest. We label Super-scaffold 85 and the other four super-scaffolds with a high proportion of co-segregating SNPs (asterisks). A close-up of these super-scaffolds is provided in Supplementary Fig. 2. **b** Proportion of biallelic variants (SNP/MNP and indels) in 200-Kb intervals with a 50-Kb step co-segregating with the Terrazzo locus in Super-scaffold 85. Dark green dots correspond to the 12 Mb region with the highest proportion of cosegregating variants. The shaded green area corresponds

to the reduced 1.9 Mb interval based on genotyping of additional individuals (Supplementary Fig. 3). An arrow points to the position of *PMEL*. **c** A nucleotide polymorphism at the end of exon 8 and the beginning of intron 8 affects the *PMEL* transcript in Terrazzo individuals (Supplementary Fig. 4). **d** InterProScan domains of the wild-type *PMEL*. The alternative Terrazzo *PMEL* proteins are 478 and 532 amino acids long (red vertical lines). The red arrows point to the target sites of the gRNAs used for gene-editing with CRISPR-Cas9. The range of the two fragments, Mα and Mβ, after cleavage are shown below. IP signal IP, CAF core amyloid fragment, PKD polycystic kidney disease domain, RPT repeat domain, CS proprotein convertase cleavage site, KLD Kringle-like domain, TM transmembrane helix, cyto cytoplasmic domain.

at the N-terminus (IP; aa 1–20), the core amyloid fragment (CAF; aa 152–227), the polycystic kidney disease domain (PKD; aa 238-299), the repeat domain (RPT; aa 306–510), the Kringle-like domain (KLD; aa 559–609), a transmembrane helix (TM; aa 638–660), and a cytoplasmic domain (cyto; aa 661–713)[21,23–25] (Fig. 2d). The CAF and the RPT domains are not identified by InterProScan, but they have previously been characterised in the corn snake PMEL[25]. Cleavage at the proprotein convertase site (CS; aa 506–511) produces the lumenal Mα fragment, which is disulphide-linked to the membrane-integrated Mβ fragment via the KLD domain. The amyloid core is produced by further fragmentation of the Mα fragment[26], with the interaction of the CAF and the PKD domains. The RPT domain is essential for proper amyloid morphology to maintain the sheet architecture[25].

Two *PMEL* isoforms have been annotated on the corn snake genome (XM_034412587 and XM_034412588) that differ by three nucleotides; the first is 3135 bp long and the second 3132 bp. In the second isoform, the three first nucleotides of exon 8 are missing. To verify how the multi-nucleotide polymorphism identified in Terrazzo impacts the transcript, we sequenced the *PMEL* transcript from one homozygous and one heterozygous Terrazzo individual (Supplementary Fig. 4a) using mRNA extracted from the dorsal skin of embryos at embryonic day 20 (E20; corresponding to the developmental stage 7 described in the Supplementary text). We identified three isoforms from the homozygous individual: (i) isoform TZ1 missing exon 8, (ii) isoform TZ2 with a 20-nucleotides insertion between exons 8 and 9, which corresponds to the first 20 nucleotides of intron 8, and (iii) isoform TZ3 with a 233-nucleotides insertion between exons 8 and 9, which corresponds to the first 233 nucleotides of intron 8 (Supplementary Fig. 4b). The three isoforms produce truncated PMEL proteins —the TZ1 protein is 478 aa long and the TZ2 and TZ3 proteins are 532 aa long—that lack the Mβ fragment, which includes the Kringle-like domain and the transmembrane helix. We cannot exclude the

presence of other isoforms. We could only sequence the WT isoform from the heterozygous individual, but the other isoforms are likely to be expressed at lower levels not detectable with our amplification and sequencing method.

## *PMEL* expression is reduced in Terrazzo corn snakes

Next, we performed differential expression analyses by bulk RNA sequencing to verify the expression levels of *PMEL* during development. We extracted RNA from the skin of three homozygous (*tz/tz*) and three wild-type (+/+) embryos at E20 (developmental stage S7; Supplementary text). This is the earliest stage at which the skin can be dissected, and early-stage unpigmented chromatoblasts—progenitors of melanophores, xanthophores and iridophores—are already present in this tissue. Note that the six embryos were from the same cross (*tz/+* x *tz/+*) and clutch to minimise expression differences due to a variable genetic background and developmental stage (Supplementary Fig. 5a, b). The most significantly downregulated gene is *PMEL* (adjusted *p* value: $4.2326 \times 10^{-165}$; log2 fold-change: −3.87; Fig. 3a). Among the 12 genes with an absolute log2 fold-change greater than 1, only *PMEL* is within the genomic interval of the Terrazzo mutation (Fig. 3b). Transcripts per million (TPM) calculations at the exon level showed that all exons are less expressed in Terrazzo compared to the wild-type (Supplementary Fig. 5c). Thus, not only the structure, but also the expression level of *PMEL* is affected in the developing skin. There are three possibilities for this observation: (i) an additional undetermined mutation of a *PMEL* regulatory element alters its expression levels, (ii) the RNA surveillance mechanism is activated by the premature STOP codon and initiates the rapid degradation of the mutated *PMEL* mRNA[27] by nonsense-mediated mRNA decay, and (iii) the number of cells expressing *PMEL* is reduced.

*ERBB3* (Receptor tyrosine-protein kinase erbB-3), a gene involved in the differentiation of metamorphic melanophores in zebrafish[28], is

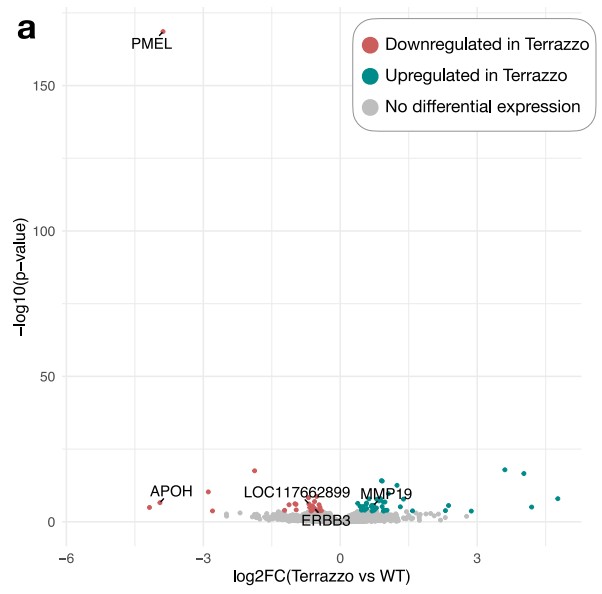

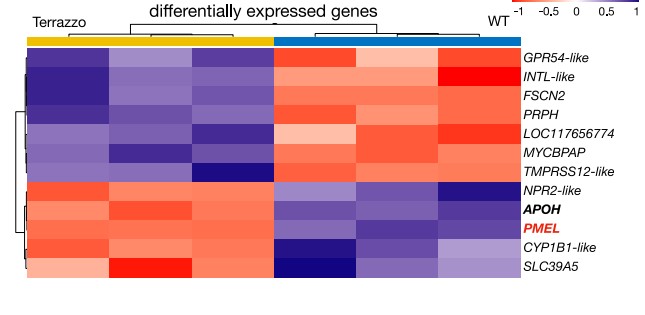

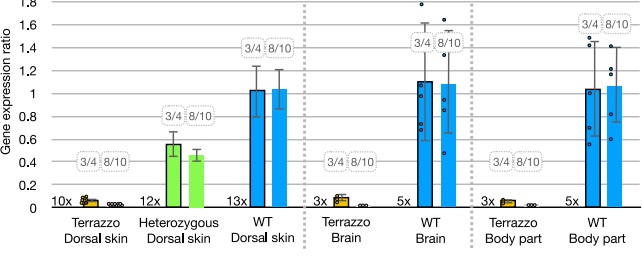

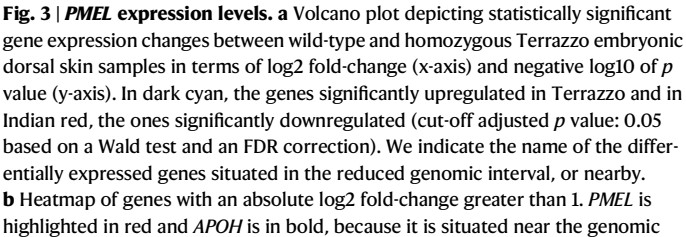

**Fig. 3 | *PMEL* expression levels. a** Volcano plot depicting statistically significant gene expression changes between wild-type and homozygous Terrazzo embryonic dorsal skin samples in terms of log2 fold-change (x-axis) and negative log10 of *p* value (y-axis). In dark cyan, the genes significantly upregulated in Terrazzo and in Indian red, the ones significantly downregulated (cut-off adjusted *p* value: 0.05 based on a Wald test and an FDR correction). We indicate the name of the differentially expressed genes situated in the reduced genomic interval, or nearby. **b** Heatmap of genes with an absolute log2 fold-change greater than 1. *PMEL* is highlighted in red and *APOH* is in bold, because it is situated near the genomic

interval. **c** Gene expression ratio of *PMEL* in the embryonic dorsal skin, the brain and a body part using two different primers sets, the first spanning exons 3 and 4 (left bar plot of each pair with a black border) and the second exons 8 to 10 (right bar plot of each pair without a border). The *PMEL* expression for the Terrazzo samples is barely detectable ($2^{-(\Delta\Delta Ct)} < 0.0003$), when using the second set of primers spanning the mutated site. The number of individuals used for each category is indicated next to the corresponding bars. Data are presented as mean ± s.d. Source data are provided as a Source Data file.

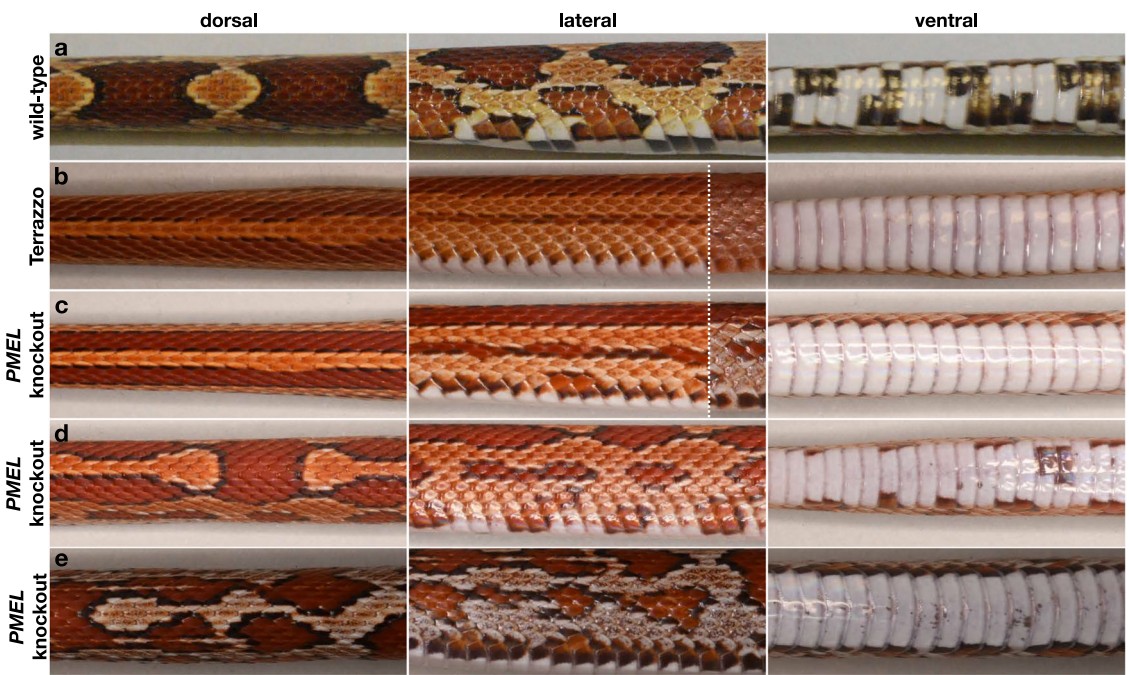

**Fig. 4 | Gene-editing of *PMEL* results in stripes instead of blotches.** Dorsal, lateral and ventral views of **a** a corn snake with a wild-type phenotype, **b** a corn snake with a Terrazzo phenotype, **c** a *PMEL* mutant with a striped phenotype, **d** a *PMEL* mutant with stripes at the anterior part of the body and blotches at the posterior and **e** a *PMEL* mutant with blotches that are not properly formed. In the lateral view of (**b**, **c**), we added photos of the posterior part of the animals (on the right of the vertical dashed lines), because the pattern changes compared to the anterior part. All animals were four months old when imaged.

present in the interval, but we do not detect any amino acid modifications in its coding sequence (Supplementary Table 2). It is significantly downregulated at E20 (adjusted *p* value <0.05), but the log2 fold-change and its expression level are very low (Fig. 3a and Supplementary Fig. 5d). Also, the *Apolipoprotein H* (*APOH*) gene is located near the interval, and it presents a similar expression profile to *ERBB3* (Fig. 3a and Supplementary Fig. 5d). As these two genes are within or near the genomic interval, we cannot exclude their involvement in the Terrazzo phenotype, despite their low levels of expression both at E20/S7 and at E25/S9 (Supplementary Fig. 5d, e).

To confirm the results of the bulk RNA-seq analyses, we also performed real-time quantitative PCR on a greater number of individuals and from different embryonic tissues (Fig. 3c). We extracted RNA from the skin of 27 E20 embryos produced by three different *tz/+* x *tz/+* crosses. We used two sets of primers, the first spans exons 3 to 4 and the second exons 8 to 10. With the first set of primers, we observe only 5.5% PMEL expression in homozygous (*tz/tz*) individuals compared to the wild-type ones (+/+) and 55% in the heterozygous. With the second set of primers, practically no expression (0.006%) is observed in the homozygous samples, because the forward primer spans the 8/9 exon boundary, where the MNP is situated. To verify if the reduced levels of *PMEL* expression is specific to the skin, we also analysed RNA from the brain and from a body piece consisting of connective tissue, developing muscles and bones (no skin and no internal organs) of eight additional individuals (three *tz/tz* and five *tz/+*). We observe the same tendency; lower levels of *PMEL* expression in the brain and the body of homozygous animals compared to the wild-type (Fig. 3c). There are two possible and non-exclusive explanations for these observations: (i) as previously postulated, the presence of an additional undetermined regulatory mutation, and (ii) the reduced number of cells expressing *PMEL*.

## CRISPR-Cas9-mediated inactivation of *PMEL* in snakes

Disruptive and missense mutations of *PMEL* have been reported in numerous animals (for example, mouse[29], horse[30], dog[31], chicken[32], and zebrafish[33]) and, in general, result in hypopigmentation, altered colouration patterning and defective melanosome biogenesis. Our genomic and transcriptomic analyses imply the presence of structural and possibly regulatory mutations affecting the Terrazzo *PMEL*. We thus proceeded to functional analyses and, using our established gene-editing protocol in corn snakes[34], we generated *PMEL* knockouts. In short, we selected two gRNAs that successfully produced *PMEL*-mutated corn snake cultured fibroblast cell lines by targeting exons 2 (204f) and 3 (302r) of *PMEL*, respectively (Fig. 2d). The gRNAs were separately injected into the oocytes of three wild-type females (two with 204 f and one with 302r), which were crossed with wild-type males. In total, we injected 59 oocytes and the females laid 42 eggs, 30 of which hatched to give 19 mutated animals (Supplementary Table 3 and Supplementary Fig. 6). As was the case for the *EDARADD* mutants[34], most induced mutations impacted both the paternal and the maternal alleles. Only one individual is heterozygous with a different mutation on each allele. The characterisation is performed on gDNA extracted from skin sheds, which is of epidermal origin, so we cannot exclude genetic mosaicism in other tissues.

Nine mutants have a wild-type skin colouration phenotype (Fig. 4a). The induced mutations in these individuals minimally impact the coding sequence of *PMEL* (e.g. insertion/deletion of one or two amino acids). The remaining ten carry disruptive mutations (e.g. premature STOP codons), and their skin phenotype is affected. Of these, seven mutants present dorsal and lateral stripes, instead of blotches, and completely lack the black checkers on the ventral side, a phenotype that strongly resembles that of Terrazzo (Fig. 4b, c). Another individual presents stripes at the anterior part and blotches at the posterior part (Fig. 4d), whereas the other two have modified blotches (Fig. 4e). These three individuals carry similar mutations as the ones with stripes. Sequencing DNA from skin biopsies taken at different regions from these animals supports non-mosaic status for this tissue (Supplementary Fig. 6). Their variable phenotype might be due to their genetic background, undetected off-target mutations, or variation in the expression levels of *PMEL* during their development. Note that the

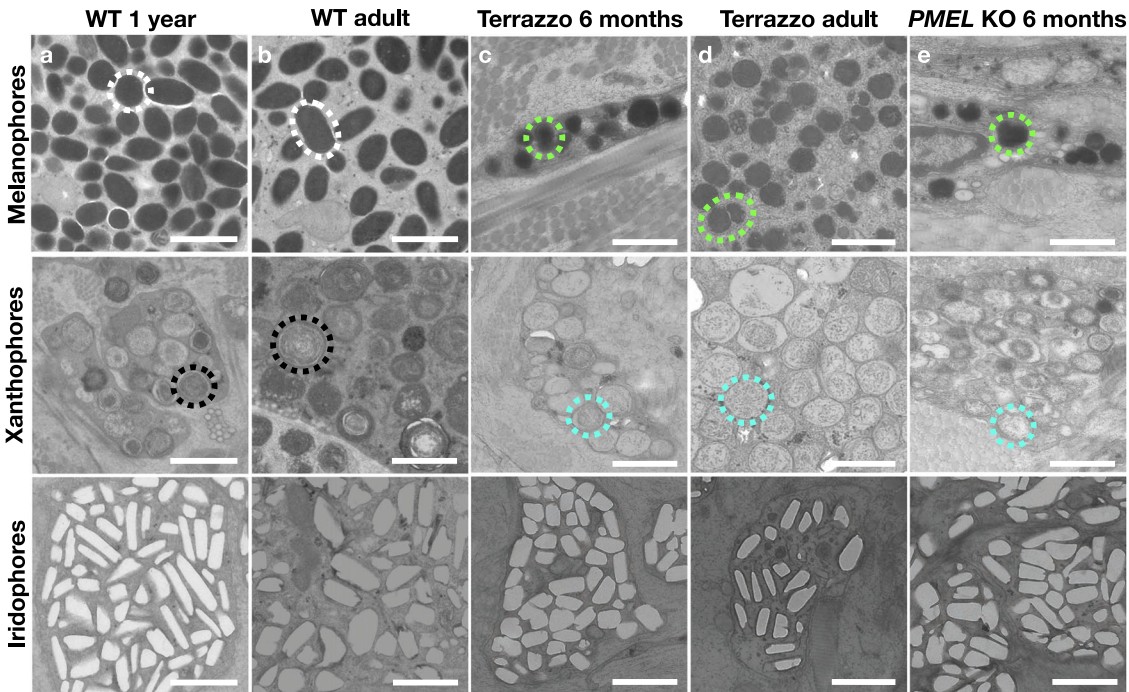

**Fig. 5 | TEM imaging reveals subcellular differences among wild-type, Terrazzo and *PMEL* knockout corn snakes.** Micrographs of melanophores, xanthophores and iridophores from the dorsal skin of **a** a wild-type juvenile, **b** a wild-type adult, **c** a Terrazzo juvenile, **d** a Terrazzo adult and **e** a *PMEL* knockout (KO) juvenile. White and green dashed circles highlight normal and modified melanosomes, respectively. Black and cyan dashed circles highlight xanthosomes with concentric lamellae and amorphous material, respectively. Three blocks of each type of sample were prepared. Scale bars: 1 μm.

cells sampled with the biopsies are not the ones that established the pattern during development.

We observed subtle differences between the *PMEL* mutants and the Terrazzo animals. The dorsal and lateral stripes of the *PMEL* mutants are clearly delineated by black lines, whereas the black border is reduced or even absent in Terrazzo animals. This difference could be due to the hypomelanistic background of the Terrazzo individuals. Indeed, the Terrazzo mutation first appeared in 'rosy rat' corn snakes, a hypomelanistic morph that exists in nature. As it has already been shown in the zebrafish[1], several genes are expected to be involved in the patterning of corn snakes chromatophores during development, and *PMEL* is only one of them. Furthermore, it remains to be seen if, over the years, the colouration of the *PMEL* mutants will become speckled at the posterior part, as is the case for the Terrazzo adults. It will also be necessary to check the expression levels of *PMEL* in the offspring of the *PMEL* mutants once available (generation time of 4 years).

### Subcellular modifications in the chromatophores of Terrazzo and *PMEL* mutants

The causative mutation of the *fading vision* (*fdv*) zebrafish mutant is a premature STOP codon in PMEL. The predicted truncated protein lacks part of the repeat domain, the Kringle-like domain, the transmembrane helix, and the cytoplasmic domain[33]. This modification closely resembles the one predicted for the Terrazzo PMEL. The *fdv* zebrafish are characterised by a reduced number of melanophores, whose amount of pigmentation and subcellular morphology is affected. Indeed, electron microscopy imaging showed that the melanosomes of *fdv* zebrafish are smaller, compared to the zebrafish wild-type melanosomes, and at variable stages of maturation. Our transmission electron microscopy (TEM) imaging revealed that the subcellular structure of both melanophores and xanthophores is greatly impacted in Terrazzo individuals from an early age compared to the wild-type

(Fig. 5a–d). Indeed, melanosomes are smaller with an irregular, instead of elliptical, shape and the xanthosomes are filled with amorphous material, rather than concentric lamellae. Note that the melanosomes take their elliptical shape only when the amyloid fibrils formed by PMEL are properly assembled. Due to the irregular shape and variable size of the guanine crystals in the wild-type, it is not possible to evaluate if iridophores are impacted as well in the Terrazzo. However, no obvious changes in their arrangement and density were observed. The subcellular structure of the melanophores is also affected in the *PMEL* mutants with small melanosomes of irregular shape (Fig. 5e). Some xanthosomes seem to contain concentric lamellae, but most of them only have amorphous material.

In the wild-type, there are melanophores and xanthophores within the blotches, and xanthophores and iridophores in the background with only scarce melanophores[19]. In the adult Terrazzo skin, we found scarce epidermal and dermal melanophores and numerous xanthophores within the stripes, whereas only epidermal melanophores and iridophores were present in the background (Supplementary Fig. 7a, b). Xanthophores are most likely present in the background as well, but in small numbers. Based on this arrangement of the chromatophores, we assume that the Terrazzo stripes roughly correspond to the wild-type blotches. At the posterior part of the body with speckled colouration, all types of chromatophores are present (Supplementary Fig. 7c). In the *PMEL* mutant, we also mainly find xanthophores in the stripe and iridophores in the background (Supplementary Fig. 8).

### *PMEL* is expressed by all chromatophore progenitors

Based on single-cell RNA sequencing experiments in zebrafish, *pmela*—one of the two *pmel* copies in this species—is mainly expressed in melanophores and their progenitors[5]. To investigate if this is the case for corn snakes, we performed single-cell RNA sequencing (Fig. 6) on dissociated cells from the dorsal skin of a wild-type S9 embryo (at E25).

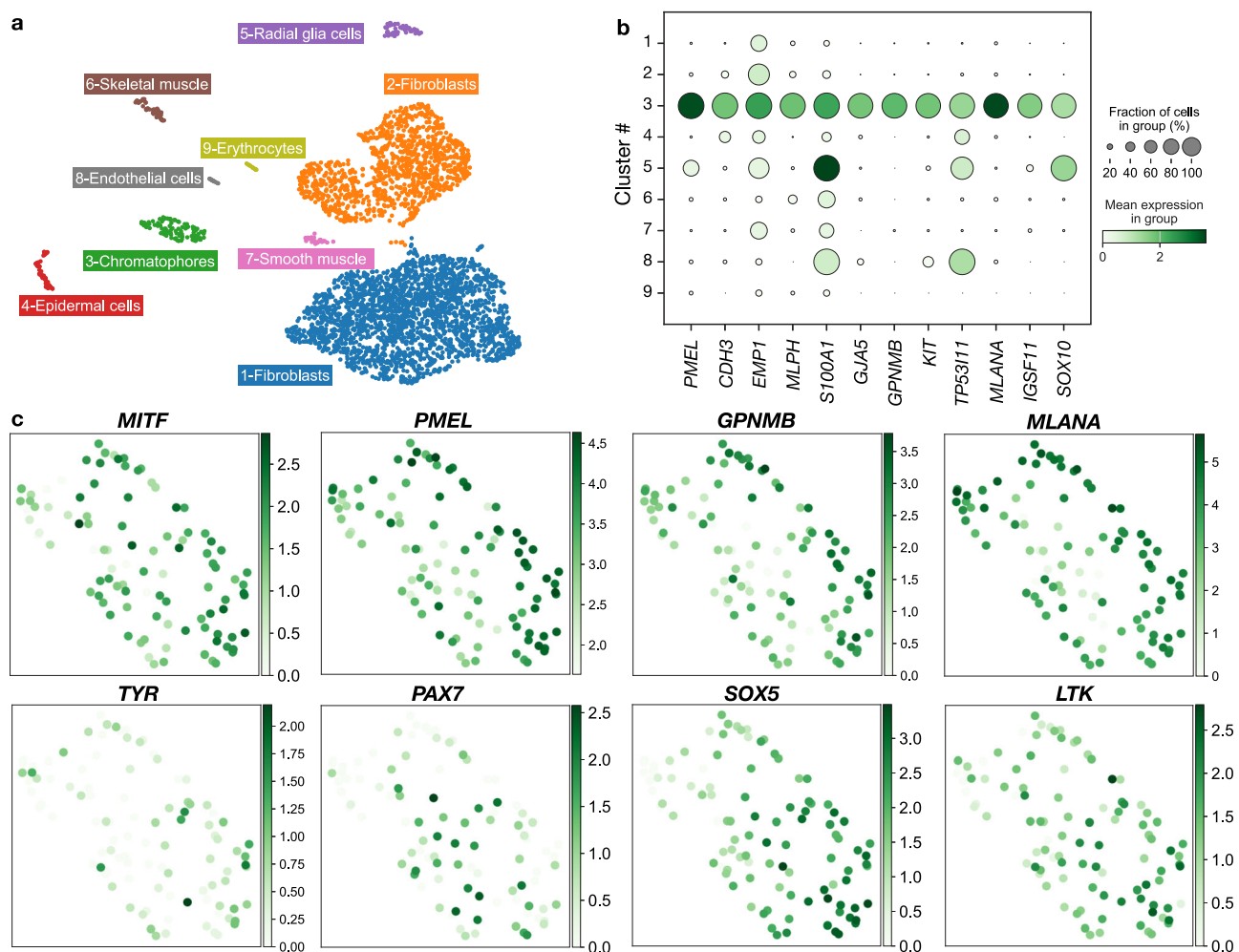

**Fig. 6 | Single-cell RNA sequencing analysis of the developing corn snake skin. a** Cell-type assignment of the nine clusters produced by unsupervised clustering. **b** Dot plot displaying the mean expression of the top 12 genes of the chromatophore cluster in all clusters, as well as the fraction of cells expressing them in each cluster. **c** Gene expression levels of selected genes in the cells of the chromatophores cluster.

We staged the embryos as described in the Supplementary text and based on the staging of African house snake embryos[35]. Although no pigmentation is visible on the S9 embryo, the development of the skin is well advanced with scale placodes formed and the patterning of the chromatophores underway (Fig. 7). After quality check, pre-processing, and filtering, we could analyse the transcriptome of 3570 cells (Supplementary Fig. 9a, b). We clustered the cells in an unsupervised manner, and we identified nine distinct cell clusters, two of which comprise 91.3% of the cells (Fig. 6a). The expression of collagen genes by these cells suggests that they belong to two fibroblast populations (Supplementary Fig. 9c). Indeed, comparison with early mouse skin markers of fibroblasts[36,37] (Supplementary Fig. 9d, e) showed that a significant proportion of upregulated transcripts in cluster 1 overlap with markers of the upper dermis and dermal condensate fibroblasts. The significantly upregulated transcripts in cluster 2 correspond to markers of fascia-forming and pre-adipocyte markers, as well as to markers of muscle-supportive fibroblast. Based on the differentially expressed genes of the remaining clusters (Supplementary Data 1), we could identify them as: chromatophores (*SOX10* and *PMEL*), epidermal cells (*keratin, type I cytoskeletal 24-like gene* (*LOC117664631*) and *keratin, type II cytoskeletal 5-like gene* (*LOC117662708*)), radial glial cells (*SOX10, FABP7* and *SOX2*), skeletal muscle cells (*MYOD1*), smooth muscle cells (*ACTA2*), endothelial cells (*TEK*), and erythrocytes (*haemoglobin subunits alpha-A, alpha-D, beta-1* and *beta-2*)[38].

Among the top 12 upregulated genes in the chromatophores cluster, we find *SOX10* (*SRY-Box Transcription Factor 10*), which is also upregulated in glial cells, reflecting the neural crest origin of both cell types (Fig. 6b). Most of the other genes are predominantly expressed in the chromatophores cluster and they are known to be involved in melanophore development and physiology, for example, *CDH3* (*P-cadherin*), whose expression promotes intrafollicular melanogenesis[39], *MLPH* (*Melanophilin*), which participates in melanosome transportation[40], and *IGSF11* (*Immunoglobulin Superfamily Member 11*), that is required for the migration and survival of melanophores[41]. Only *S100A1* (*S100 Calcium Binding Protein A1*) and *TP53I11* (*Tumour Protein P53 Inducible Protein 11*) show a wider expression in different clusters. *S100A1* was previously shown to be expressed in the brain and in skeletal and heart muscles, where it is involved in cardiac contractility regulation[42]. *TP53I11*, a target of TP53, has a tumour-suppression function[43] and a ubiquitous expression[44,45].

Within the chromatophores cluster (Fig. 6c), we observe a broad expression of *MITF* (*Melanocyte Inducing Transcription Factor*), which is consistent with previous findings in the zebrafish[5]. Indeed, in zebrafish, *mitfa* is expressed by all pigment progenitors and persists in subpopulations of melanophores and xanthophores[5], despite that it is only required for melanophore fate specification[46]. Our previous findings in a closely related species, the Texas rat snake, suggest that, in snakes, *MITF* plays a role in the differentiation of both melanophores

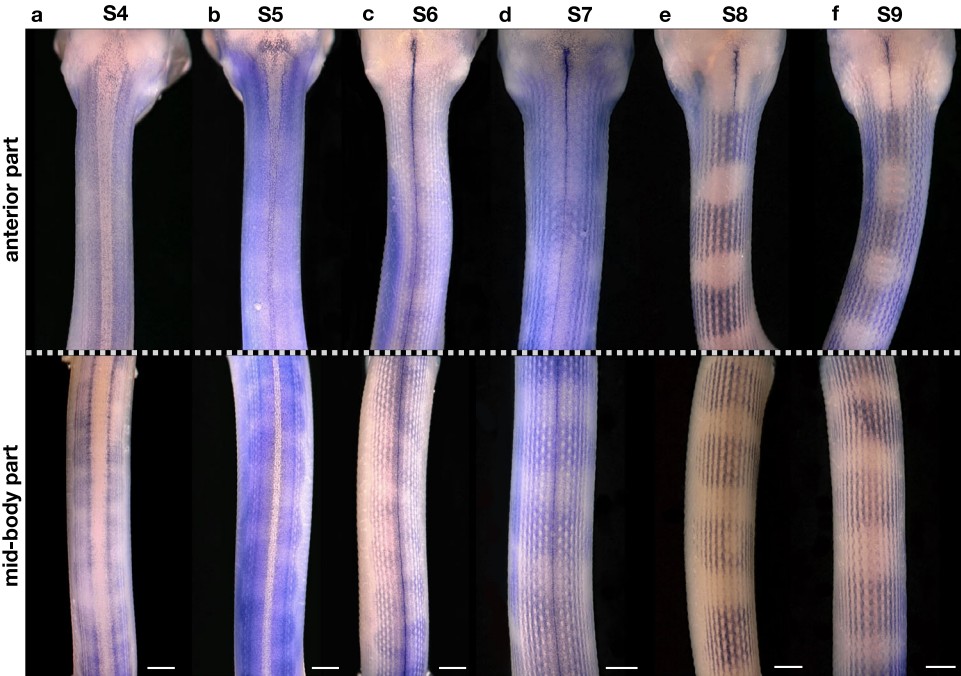

**Fig. 7 | Whole-mount in situ hybridisation of wild-type embryos with a PMEL probe.** Dorsal view of the anterior and mid-body part of embryos at developmental stages **a** S4, **b** S5, **c** S6, **d** S7, **e** S8 and **f** S9. The stages are described in the Supplementary text. At least two embryos were prepared for each stage. Scale bars: 1 mm.

and xanthophores[47]. Indeed, *MITF* Texas rat snake mutants are leucistic, lacking melanophores and xanthophores, but maintaining iridophores. We thus assume that at this stage of development, the chromatophores cluster mainly consists of progenitors at early stages of differentiation. *PMEL* is expressed by all chromatophores at varying levels. These progenitors also express early melanophore differentiation markers *GPNMB* (*Glycoprotein Nmb* – 95.1% cells), and *MLANA* (*Melan-A* – 94.12% cells). On the other hand, late differentiation melanophore markers, such as *TYR* (*Tyrosinase*), are expressed by a small subset of cells and in low levels. Similarly, only a few cells express xanthophore and iridophore determination markers, such as (i) *PAX7* (*Paired Box 7*), involved in zebrafish xanthophore differentiation[48], (ii) *SOX5* (*SRY-Box Transcription Factor 5*), which is implicated in xanthophore specification in zebrafish and leucophore specification in medaka[49], and (iii) *LTK* (*Leucocyte Receptor Tyrosine Kinase*), expressed by zebrafish iridophores and their precursors[50]. Presumably these cells have committed to the xanthophore and iridophore lineages, despite still expressing melanophore markers. Note that no pigmentation is visible at the developmental stage we sampled. Melanin production only starts after E32, and the iridophores and xanthophores fully mature after E40.

## Spatio-temporal expression pattern of *PMEL* in developing snakes

Next, we performed whole-mount in situ hybridisations (WISH) using a *PMEL* probe to elucidate its expression pattern in the developing skin. In the wild-type, *PMEL* is first expressed at S3 in individual cells that form two dorsal stripes (Supplementary Fig. 10a). At this stage, the skin over the neural tube is very thin and the coelum is open ventrally, so we consider that the left and right sides of the embryo develop independently[34]. At S4 and S5 (Fig. 7a, b), a greater number of cells express *PMEL*. At the anterior part near the head, the skin over the neural tube becomes thicker and is populated by labelled cells. At the mid-body part, the labelled cells start to form aggregates, which are not aligned on the left and right sides. At S5, scale placodes become apparent, as regions devoid of labelled cells. At S6 (Fig. 7c), the skin over the neural tube is fully developed, and the scale placodes cover

the embryonic skin entirely. At this stage, the aggregates of *PMEL*-positive cells are not easily discernible, possibly due to the staining. From S7 to S9 (Fig. 7d–f), the aggregates of labelled cells gradually become more defined and finally resemble the dorsal blotches of the adult (Fig. 1a), with the alignment of the left and right side. Note that labelled cells can also be found in the background, around the future blotches, but their concentration is reduced. Finally, we observe that at these stages of development, the chromatophore progenitors avoid the regions of the scale placodes, in a similar manner as chromatophores avoid the developing lateral line in salamanders[51]. We also performed WISH on a developmental series of Terrazzo embryos, and we were not able to detect any cells expressing *PMEL* (Supplementary Fig. 10b, c), despite that the probe spans exons 2 to 6, an intact region of the Terrazzo *PMEL* transcript. The *PMEL* expression, if any, is likely below the detection threshold of the hybridisation staining in the Terrazzo individuals.

To elucidate the impact of the *PMEL* mutation on the number of chromatophores in Terrazzo, we performed WISH with an *MLANA* and a *GPNMB* probe, because they are expressed by the majority of the chromatophores at similar levels as *PMEL* based on our single-cell transcriptomic analyses (Fig. 6c). Thus, if the number of chromatophores remains the same in Terrazzo, we expect the expression levels of *MLANA* and *GPNMB* to be similar in wild-type and Terrazzo individuals, independently of the patterning process. We rather observe a very low staining with probes for both genes in Terrazzo (Fig. 8), at the regions where the dorsal stripes will eventually form. Although accurate quantification is not possible with WISH, in the higher magnification images, we see that the number of positive cells is lower in Terrazzo, suggesting a reduction in the number of chromatophores. Note that reduced expression of these markers in Terrazzo individuals is also confirmed by the bulk RNA-seq data (Supplementary Fig. 5f). These observations suggest an overall reduction of the chromatophore cells in Terrazzo, rather than the presence of an undetermined regulatory mutation that would exclusively affect the expression of *PMEL*. Furthermore, it seems like the patterning process is arrested in Terrazzo individuals and the aggregates and blotches do not form, thus the presence of stripes.

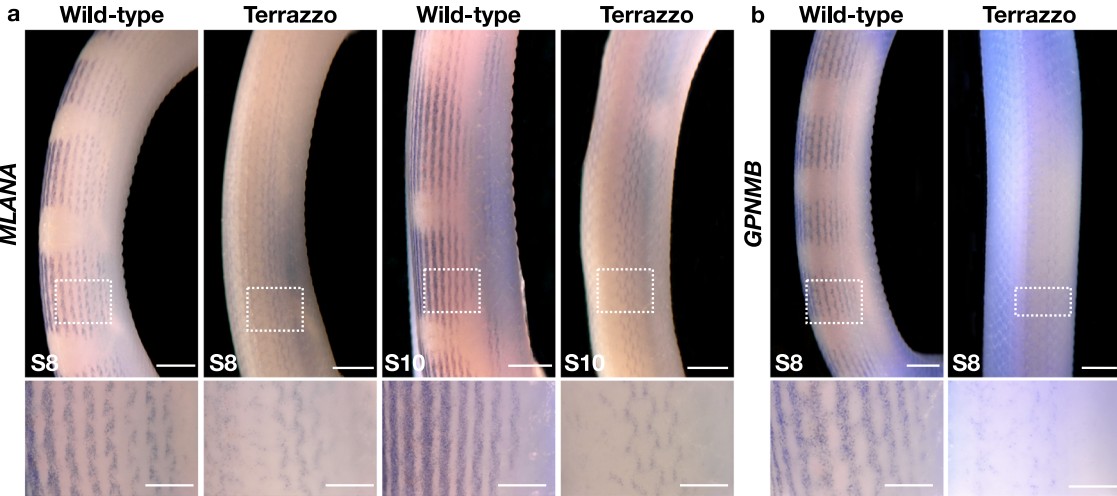

**Fig. 8 | Whole-mount in situ hybridisation of wild-type and Terrazzo embryos with *MLANA* and *GPNMB* probes.** Dorsolateral view of the mid-body part of wild-type and Terrazzo embryos at different developmental stages on which we performed WISH with *MLANA* (**a**) and *GPNMB* (**b**) probes. The regions in the dashed rectangles are magnified at the bottom. At least two embryos were prepared for each stage. Scale bars: 1 mm; magnified areas: 0.5 mm.

Based on the single-cell transcriptomic data and the WISH experiments, we conclude that the subtle differences in the expression profile of the chromatophore progenitors seem sufficient for the early establishment of the blotched pattern (Fig. 7). It is likely that most cells expressing *PMEL* at a detectable level form blotches, similarly to the adult pattern. The expression of *PAX7*, a classical xanthophore marker, is barely detectable by whole-mount in situ hybridisations, but it seems to be mainly present in the background region (Supplementary Fig. 11a). This result should be taken with caution and highlights the need to identify species-specific xanthophore and iridophore markers, given that the markers from zebrafish studies, such as *CSF1R*[52] and *GCH2*[48] for xanthophores, as well as *TFEC*[53] and *CSF1*[52] for iridophores, are barely expressed in corn snakes at this stage of development when the pattern is established (Supplementary Fig. 11b).

## Discussion

Our genomic, bulk, and single-cell transcriptomic, functional, and developmental characterisation of the Terrazzo corn snake colour morph and the extensive comparison with wild-type corn snakes puts forward the dual role of *PMEL* in snake skin colouration, both in the patterning of chromatophores during embryogenesis and the melanogenesis in melanophores. Our subcellular imaging revealed that the Terrazzo melanophores are smaller and misshaped, probably due to the truncated PMEL protein they produce. Indeed, in the absence or modification of the Mβ fragment (including the Kringle-like domain and the transmembrane helix), it is possible that PMEL is not properly transferred inside the early-stage melanosomes. The presence of PMEL amyloid fibrils formed by the Mα fragment in the cytoplasm could be toxic for the cells. This toxicity could explain the gradual transition from a striped to a speckled pattern in adult Terrazzo animals. We were intrigued to also find modifications in the subcellular structure of Terrazzo xanthosomes, which were missing concentric lamellae. Indeed, very little is known about the formation of the lamellae in xanthosomes, and PMEL has never been associated with them. These results should be taken with caution as they could be linked to the genetic background of the Terrazzo animals, rather than the Terrazzo causative mutation. It is worth noting, though, that, besides the melanosomes, the xanthosomes of the *PMEL* mutants we produced were also impacted. We now need to wait for these gene-edited animals to grow, such that we can verify if the subcellular structure of their chromatophores and their skin pattern will change over time.

Both the Terrazzo corn snakes and the *PMEL* mutants we generated are characterised by stripes rather than blotches, confirming the involvement of *PMEL* in the patterning process. From our single-cell experiments and developmental studies, we conclude that *PMEL* is a marker of chromatophore progenitors, which we can use to follow the pattern formation during development. In wild-type embryos, the chromatophore progenitors homogeneously populate the developing skin, while they differentiate from neural crest cells. As their numbers increase, they start to independently form aggregates on the left and right sides of the embryo. When the skin of the two sides merges, the aggregates align, and the future blotches become apparent. Note that we also observed such a uniform distribution of *PMEL*-expressing cells during the establishment of the leopard gecko skin pattern[14], before the bands of the hatchlings are formed. In Terrazzo embryos, expressing low levels of mutated *PMEL* isoforms, we speculate that the patterning process is arrested at an early stage, when the *PMEL*-expressing cells are evenly distributed at each side of the dorsal midline, resulting in stripes rather than blotches. It is possible that the number of chromatophore progenitors in Terrazzo is reduced and, therefore, the aggregates do not form. Thus, we postulate that a minimum number of chromatophore progenitors is necessary for the aggregates and, later on, the blotches to form.

In Nile Tilapia fish, double *pmela* and *pmelb* knockouts have a smaller number of melanophores and a greater number of xanthophores[54]. In the case of the Terrazzo corn snakes, we suspect that the number of all chromatophore progenitors is reduced. Alternatively, changes in *PMEL* could primarily impact melanophores differentiation and consequently the patterning of the other chromatophores, in a similar manner as the overexpression of *csf1a* primarily impacts xanthophores, but then the patterning of all chromatophores is modified in zebrafish[55]. It is possible that when removing the dorsal skin, we primarily sample the melanophore progenitors, so sampling different parts of the embryo could reveal the presence of progenitors of the other lineages. Note that our single-cell transcriptomic analyses on the dorsal skin of a wild-type leopard gecko embryo also support a uniform expression of *PMEL* by all chromatophore precursors, before they differentiate and mature[14].

To support our hypothesis on pattern formation, we would need to perform single-cell experiments at different developmental stages and describe the dynamic expression profiles of the chromatophore progenitors as they differentiate and mature. These transcriptomic

experiments would also allow us to identify markers of each type of chromatophores and investigate their spatial distribution at each developmental stage. The establishment and optimisation of gene-editing experiments will also enable us to understand the role of key players in the colouration of other vertebrates, such as *ERBB3*, whose expression level we found low in corn snake and its function in reptilian colouration, if one, remains unknown. As we continue the characterisation of different colour and colour pattern morphs of corn snakes and other squamate model species, we will deepen our understanding of the developmental processes that bring about the spectacular diversity of their skin colouration and unravel the evolution of the molecular mechanisms responsible for animal colouration.

## Methods

### Experimentation model
Corn snakes were housed and bred at the LANE animal facility running under veterinary cantonal permit no. 1008. Sampling and imaging were performed under the experimentation permits GE24/33145 and GE150/34215.

### Crosses and deep sequencing of Terrazzo individuals
Genomic DNA was extracted from the parents and the offspring using the QIAGEN DNeasy Blood and Tissue kit (69504) following the manufacturer's instructions. The TruSeq DNA PCR Free libraries were sequenced using an Illumina HiSeqX instrument, producing 151 bp paired-end reads (Macrogen). We obtained 173.1 to 185.9 million paired-end reads per library. We checked the data quality and the presence of adaptors with FASTQC. We performed a quality filtering with sickle v1.33[56]. We retained between 167.4 and 179.8 million reads, which correspond to a 29.7-31.9x average coverage for a 1.7 Gb genome.

### Variant calling
The four genomic libraries (two for the parents and two for the offspring) were aligned to the corn snake genome[19] using bwa v0.7.16[57] with default parameters in mem mode. We used SAMtools v1.9[58] to: convert the output SAM files into BAM, to remove duplicates using the fixmate mode with the -m flag and the markdup mode with the -r flag, and to sort out the reads by their leftmost coordinates. We identified genomic variants with Platypus v0.8.1[59] and retrieved the genomic interval where the Terrazzo locus is located as previously published[19].

### *PMEL* transcript amplification and cloning
We extracted mRNA from embryonic tissue with the Direct-zol RNA Miniprep kit (Zymo Research, R2081) and prepared complementary DNA with PrimeScript™ Reverse Transcriptase (Takara Bio, RR047A). We then amplified and sequenced the *PMEL* transcript in a Terrazzo and a heterozygous individual. The primers are provided in the Supplementary Table 4.

We extracted from an 1% agarose gel the three visible bands amplified with the primers EG_PMELc_1321-F/EG_PMELc_2090-R from a homozygous individual using the QIAquick Gel Extraction Kit (Qiagen, 28704). The extracted bands were Sanger sequenced with the same primers.

### Bulk RNA-seq sampling and analysis
We first extracted genomic DNA from a piece of the tail from each E20 embryo, and we genotyped them using the primers EG_sc85-2900655-F/EG_sc85-2901018-R and EG_sc85-14891026-F/EG_sc85-14891444-R. We then extracted total RNA from the skin of three homozygous Terrazzo and three wild-type embryos using the Direct-zol RNA Mini-Prep (Zymo Research, R2050). We sequenced between 18.8 and 21 million reads from each TruSeq Stranded mRNA Library. For the skin microdissection, we carefully detached the skin along the opening of the ventral side before pulling it off. Sufficient RNA was obtained from

each embryo, so the samples were not pooled. All samples had an RNA integrity number (RIN) ≥9.6.

The bulk RNA-seq samples were aligned to the corn snake genome (GCF_001185365.1) with STAR (v2.7.0d) using default parameters for paired-end libraries. Gene expression quantification was performed using the featureCounts function implemented in the R package Rsubread (v2.2.1), counting uniquely mapped paired-end reads. The filterbyExpr function implemented in the edgeR package (v3.40.2) was used to filter out genes with a low number of counts for downstream analyses. Data normalisation, transformation (considering variance stabilising transformation), principal component analysis and differential expression analyses with the Wald test and an FDR of 0.05 were performed with the DESeq2 package (v1.38.3). To obtain the read count that maps to each *PMEL* exon with featureCounts, we generated a GTF file that included only *PMEL* with a unique ID per exon. We disregarded multimapping reads but considered overlapping reads that span more than one exon. For the TPM calculation, we corrected for the exon length and the library size.

### Real-time quantitative PCR
We extracted RNA with the Direct-zol™ RNA MiniPrep (Zymo Research, R2050). Reverse transcription starting with 500 ng of RNA was performed with the PrimeScript RT Reagent Kit with gDNA Eraser (Takara Bio) which includes a step for the elimination of any gDNA contamination. We used the PowerUp SYBR Green Master Mix (Thermo Fisher Scientific) for the quantitative RT-PCR run on a QuantStudio 5 Real-Time PCR System (Thermo Fisher Scientific). All samples were run in triplicates, and the primers are provided in Supplementary Table 4. We verified the PCR efficiency[60] for the amplification of *PMEL* and *ALAS1*, the reference gene, and performed the statistical analyses as previously described[61].

### Gene-editing with CRISPR-Cas9
General anaesthesia for the surgical procedure was induced by subcutaneous injection of alfaxalone (15 mg/kg) and medetomidine (0.1 mg/kg). The animals were then intubated and maintained on intermittent positive pressure ventilation with 3–7% sevoflurane in a mixture of oxygen and air. Reversal of the anaesthesia was accomplished by subcutaneous injection of atipamezole (0.5 mg/kg). Analgesia was accomplished in a multimodal approach: (i) pre-emptive analgesia was provided by subcutaneous injection of meloxicam (0.2 mg/kg) at the induction of anaesthesia, (ii) intraoperative analgesia was provided by the medetomidine and local infiltration of the skin incision site with lidocaine (4 mg/kg), and (iii) post-operative analgesia was provided by one daily subcutaneous injection with meloxicam (0.2 mg/kg) for 3 days. Antibiotic coverage was applied prophylactically (Ceftiofur CFA, 15 mg/kg, single subcutaneous injection). The surgical procedure to access the ovaries and inject the follicles was identical to the previously described[34]. Genomic DNA was extracted from the skin sheds of the offspring with the DNeasy Blood and Tissue Kit (69504, QIAGEN). We used the FastStart polymerase (12032902001, Sigma) to amplify the target regions. The gRNAs and the primers for the amplification of the target regions are detailed in Supplementary Table 4.

### Single-cell RNA-seq analysis
E25 corn snake skin was isolated from a single embryo and dissociated in a single cell suspension according to a protocol used for chicken embryos[62]. Briefly, the tissue was treated with a solution constituted of 0.25% Dispase II (Sigma, D4693) and 0.25% Trypsin in Hank's balanced solution (Sigma, M4780) at 37 °C for 18 min. Hank's balanced solution supplemented with 0.25% BSA (Sigma, A3294) was used for Dispase II and trypsin inactivation, cell suspension washing and resuspension. Subsequently, the single-cell RNA library was prepared using the 10X Genomics Chromium Single Cell 3' V3 kit. The library was sequenced on a HiSeq X Ten. Cellranger (v7.0.0) identified 5,431 cells. The reads

were mapped against our corn snake genome assembly (GCF001185365.1) and the corn snake mitochondrial genome[63] (AM236349.1). Further details about quality metrics, filtering thresholds and downstream analysis are provided in Supplementary Fig. 8b. The potential doublets were detected with Scrublet v0.2.3[64]. The differentially expressed genes in each cluster were identified using the scanpy rank_genes_groups method with default parameters. Expression values were normalised to a uniform depth of 10,000 read counts per cell using the scanpy normalize_per_cell function and subsequently underwent a logarithmic transformation using the log1p function.

### Whole-mount in situ hybridisations

We designed species-specific digoxigenin-labelled antisense riboprobes for *PMEL*, *MLANA*, *GPNMB*, and *PAX7* (Supplementary Table 4). Embryos at different developmental stages were fixed in 4% paraformaldehyde and dehydrated in methanol. Whole-mount in situ hybridisations (WISH) were performed as previously described[65]. Embryos were imaged using the VHX-6000 (Keyence).

### Transmission electron microscopy

Skin pieces of 1 mm$^2$ were fixed and sectioned as previously described[19]. Micrographs were taken with a transmission electron microscope Philips CM100 (Thermo Fisher Scientific) at an acceleration voltage of 80 kV with a TVIPS TemCam-F416 digital camera (TVIPS GmbH). Large montage alignments were performed using the Blendmont command-line from the IMOD software[66]. The display levels of the images were adjusted with ImageJ to facilitate comparisons.

### Reporting summary

Further information on research design is available in the Nature Portfolio Reporting Summary linked to this article.

## Data availability

The sequencing data generated in this study have been deposited in NCBI under accession code PRJNA1073516 for the genomic sequencing, GSE262160 for the single-cell and GSE262159 for the bulk RNA sequencing data. Source data are provided with this paper.

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

## Acknowledgements

We would like to thank Ingrid Burgelin, Carine Langrez, Adrien Debry and Florent Montange for their technical assistance. Jean Daraspe, Antonio Mucciolo and Rodrigue Peraldi participated in the TEM sample processing and imaging, which was performed at the Electron Microscopy Facility, University of Lausanne (Switzerland). The computations were run on the Baobab cluster of the University of Geneva (Switzerland). AUA received an iGE3 PhD award and the Sara Borrell grant (CD22/00027) from the Instituto Carlos III and NextGenerationEU. Funding was provided to A.C.T. by the SNF (grant 310030_204466), the HFSP (RGP0037/2022), the Ernst and Lucie Schmidheiny Foundation (10_2023), the Fonds Général de l'Université de Genève (23_28) and the Emile Plantamour Fund (2024/12).

## Author contributions

Conceptualisation, funding acquisition, methodology, project administration, supervision, validation and writing—original draft: A.C.T. Data curation and visualisation: A.C.T., A.U.-A. and P.-Y.H. Investigation and writing—review and editing: A.C.T., A.U.-A., P.-Y.H. and M.K. Software: A.U.-A. and P.-Y.H.

## Competing interests

The authors declare no competing interests.
