## [Peer Review File · Nature Communications]

PMEL is involved in snake color pattern transition from blotches to stripesREVIEWER COMMENTS

Reviewer #1 (Remarks to the Author):

In this study Tzika et al. identify the gene PMEL as corresponding to the domesticated corn snake morph, Terrazzo, by genetic mapping and sequencing, with supporting evidence from CRISPR knockout and gene expression analyses. The authors further demonstrate the predominant expression of PMEL in chromatophores by scRNA-seq and they examine roles for PMEL in chromatophore development by comparing gene expression in wild-type and Terrazzo/PMEL mutants by in situ hybridization and chromatophore ultrastructure by TEM. The study is thorough and the text is clear and I have only minor comments for the authors to consider.

1. The comparison of PMEL F0 knockout phenotypes to Terrazzo and other variants could be revised. Though the authors suggest F0s are closer in phenotype to Terrazzo than other variants, the relative degrees of similarity seem a bit ambiguous, at least to this reviewer. The important point here is that the PMEL F0 phenotypes have similarities to that of Terrazzo and so help to establish the correspondence of Terrazzo to PMEL. Since mutations in different genes can often give the same phenotypes, anyway, showing the other variants here, and inferring similarities or differences, does not really add new information relevant to the Terrazzo/PMEL gene identification per se. In combination with inferences on PMEL function (from ISH), however, it is potentially interesting that similarities exist between F0s and other variants, as they would suggest a common mechanism at the cellular level (i.e., a progenitor deficiency in the authors' model). The authors should also clearly state whether the other variants are known to be at loci other than Terrazzo from breeder data (i.e., complementation tests, co-segregation analyses), and if somewhat uncertain one might wonder about sequencing PMEL from them to check.

2. The variation in F0 phenotypes is intriguing. Were these zebrafish for example, one would assume that differences among individuals and relative to the original variant reflect underlying mosaicism of the cells involved. Yet the sequencing is said to have revealed homozygosity in most instances with mutation even to the paternal allele post-fertilization. Given the ambiguity I recommend showing some Sanger electropherograms as examples, as many readers will be used to looking at these in the context of other species for which they might be analyzed computationally to reveal the degree of mosaicism. Additionally, it seems important to assess whether there is indeed any mosaicism in the relevant population of cells, as one could easily imagine that phenotypic heterogeneity reflects clonal expansion of small numbers of escaper cells, which--as the authors carefully point out--might not be evident in shed epidermis. Of course the animals are precious given the difficulty of generating them, and presumed regulatory constraints on using them. But if sequencing of small dermal biopsies were possible it would potentially be informative for interpreting these phenotypes and those of other species in which biallelic knockout has been seen.

3. At lines 320 and 326 it appears that Figure 6 should be cited; at line 358, suppl figure 8C.

4. Imaging of progenitors stained by IHC really would benefit from higher magnification details,

sufficient to show individual cells rather than broad (and seemingly diffuse) pattern. Since the authors have these specimens presumably, quantitative assessments of cell numbers in defined regions would seem possible to obtain. These would strengthen the conclusions and would align with the manuscript text that seeks to assess determine the impact "on the number of chromatophores."

5. At line 281, a brief description of stage S9 would be helpful (e.g. relative to NC migration, pigment cell differentiation). It appears later but the reader could use the information up front in evaluating scRNA-seq results.

6. In suppl figure 8, add annotation to indicate Terrazzo for the lower panels.

7. In figure 4, add to legend a description of the vertical dashed line and image splice/comparisons for b and c lateral views.

8. At line 35, zebrafish might be considered a reference for some fishes, or perhaps ectotherms (a stretch), but not animals; similarly, suggest avoiding at line 374 the term "classical" based on zebrafish markers. As the authors' group recognizes, and is demonstrating nicely, we just don't know enough about other clades or even other fishes. I'd let zebrafish be zebrafish.

9. Display levels of TEM in figure 5 could reasonably be adjusted to facilitate comparison of panels. The authors might note that adjustments were made for this purpose in the methods, if concerned about seeming to manipulate images.

Reviewer #2 (Remarks to the Author):

Review: Accepted with minor revisions

A major revision in interpretation is required (noted in the following paragraph). All other suggestions are minor revisions and listed below.

Summary

The authors demonstrate that PMEL is involved in the pigmentation patterning of corn snakes, and changes to PMEL isoforms and/or expression can lead to modified pigment patterns. This result has not been demonstrated before, and it is a noteworthy and interesting result. This work is likely of significance to understanding pigmentation pattern development, as it contributes to our understanding of how PMEL may contribute to pattern development. However, the authors do not

address how the work may be of significance to other fields.

Overall, the approach to determining the potential mechanism by which blotches become stripes in corn snakes is very methodical. They take a top-down approach (mapping-by-sequencing) that allows a relatively unbiased identification of candidate genes, and then followed the most likely candidate (PMEL). They tested the phenotypic effect of this candidate using a knockout mutant and other follow-ups to begin understanding the mechanisms underlying these corn snake pigmentation patterns.

Their first claim of a disruptive mutation (premature stop codons) in PMEL is present in Terrazo was supported in a single individual, but not necessarily for all Terrazo individuals. However, they showed strong and significant down-regulation of that disrupted exon and PMEL generally in multiple Terrazo individuals, indicating that PMEL is a strong candidate driver of pigmentation patterning in corn snakes despite the earlier low sample size. The authors generated PMEL knockouts using CRISPR-Cas that qualitatively have similar subcellular pigmentation structures to Terrazo mutants. Using single-cell RNA sequencing, the authors conclude that PMEL is expressed to varying levels in chromatophore progenitors; furthermore, they state that chromatophore progenitors were reduced in Terrazo individuals. I believe that these claims around the chromatophore progenitors require major revision in interpretation. First, the authors did not perform cell counts on chromatophores or chromatophore progenitors in Terrazo and/or knockout individuals. Their claim of reduced numbers relies on *in situ* hybridization, but their figures demonstrate changes in ontogenetic PMEL expression, not necessarily in chromatophore progenitor numbers. Second, they claim that PMEL is present in all chromatophore progenitors, but I believe that they are basing this conclusion on zebrafish chromatophore development. It is important to note that chromatophore differentiation could be different in squamates (or even specifically corn snakes) compared to teleosts. Furthermore, zebrafish chromatophore differentiation is complex, and melanophore markers decrease in certain chromatophore progenitors ontogenetically as fate specification turns them into xanthophore or iridophore progenitors (see: Subkhankulova et al. (2023) *Nature Communications* (doi: 10.1038/s41467-023-36876-4). I find myself thinking that melanophore differentiation is affected (reduced), which has in turn affected the patterning of the other pigment cells. In zebrafish, changes in *Csf1* affected xanthophore differentiation, which led to cascading effects on other chromatophores' differentiation and patterning (Patterson et al. 2014; *Nature Communications*; DOI: <https://doi.org/10.1038/ncomms6299>). Similarly, changes in PMEL could be having cascading effects on differentiation and patterning, but not necessarily on chromatophore numbers, which would require cell counting. Perhaps this section would be improved with a figure in the discussion about PMEL expression in chromatophore differentiation and development (in short, a conclusion figure that wraps up their results and/or future testable hypotheses would be great). Finally, their whole *in situ* hybridization does support PMEL-expressing cells gathering in aggregates in wild type but not Terrazo individuals, strongly suggesting that PMEL is important for the creation of the blotched patterning in wild type corn snakes.

Suggestions for minor revisions are written below, primarily asking for clarification of methods for replicability and for corrections to figure references in the main text.

Other Suggestions

Introduction

Line 51 include species names where possible

Figure 1: Beautiful photographs. I recommend either labeling dorsal and ventral on the diagram directly or specifying top/bottom in the figure legend.

Line 58 A brief description of the mapping-by-sequencing approach (or a reference to a review, such as Candela et al. (2014) *Journal of Integrative Plant Biology*) would be helpful to frame the approach.

Methods & Results

Description

In both Lines 82-99 of the main text and Lines 17-53 of the supplementary text, it is not clear at what developmental time point body pigmentation appears. Although later, the authors do mention that chromatophore maturation appears to occur after E32 and E40, but I am unclear how pattern development relates to the chromatophore maturation and the timepoints chosen for analyses throughout the article.

Mapping

Line 102: Could you provide a supplement or reference to previously published work so that others can confirm that it is a single-locus recessive mutation?

For Figures 2 and S2: I'm not sure that I understand how Super-scaffold 85 has more biallelic variants, when it seems like 41126 and 82133 both have a high proportion of biallelic variants co-segregating. I do agree that 85 has the closest to 1, which they state indicates a high proportion are co-segregating. I personally feel that I do not understand Figure 2A and 2B well enough to comment further on this graphical display.

Lines 159-160: Unclear why they could only sequence the wild type isoform from the individual. Do they mean that it's the only isoform that they could detect at high enough levels for sequencing?

For the Discussion: I would like to see the authors discuss that the premature stop codons occur before the Kringle-like domain and the transmembrane helix, as that would be important for function of PMEL.

Methods

Line 434: What age were the adults and the offspring? Could the offspring be sexed?

Expression

Figure S5A: Based on the PCA for the RNA expression profiles of each sample, one wild type individual seems to closely group with Terrazo. Is it possible that the individual is heterozygous? I am unsure how homozygous wild type and heterozygotes were distinguished, especially if they have the same pigmentation pattern (a cross, perhaps?). However, I do not see this PCA result affecting the differential gene expression analyses (DGEA), as the DGE profiles cluster by Terrazo or WT (Figure 3B).

Line 179-183: Given that ERBB3 affects melanophores, and that the black outlines around the blotches in WT corn snakes are lost in Terrazo mutants but maintained in some crispants, I'm surprised that the authors are so quick to dismiss this finding. A change in low expression levels does not mean that the phenotypic effect is also low. For example, Ruzycski et al. (2015; *Genome Biology*; DOI: <https://doi.org/10.1186/s13059-015-0732-z>) found that, although expression correlated with phenotypic severity of their disease model, the effect was based on a threshold of gene expression differences, as opposed to a linear effect – and even a minor difference had an effect. I don't necessarily expect the authors to follow up on this gene for this study, but they should update their paper (add to discussion) to reflect this possibility and consider follow-up studies on this gene.

The authors did not appear to examine gene ontology and pathway expression of the differentially expressed genes (RNASeq). They could perform an enrichment analysis to determine whether the genes are enriched within a similar KEGG pathway, potentially with a package like Enrichr (Chen et al. 2013; *BMC Bioinformatics*; DOI: <https://doi.org/10.1186/1471-2105-14-128>).

Methods

Missing information for RNASeq (following MINISEQE guidelines):

Was rRNA removed or mRNA selected for (e.g., poly-A tail selection)?

What quality checks were performed on the RNA samples (e.g., Nanodrop, Qubit, Bioanalyzer with RIN, FastQC)?

Knockout

Although the authors did identify that mutating PMEL causes changes to the pigmentation phenotype that resembles Terrazo, they only looked at crispants and not further generations of knockout mutants. Given that this limitation appears to be due to the long generation time of corn snakes, I do not believe it is necessary for the authors to provide this additional information at this time.

Methods

I found it unclear how the crispants were genotyped. Did the authors sequence PMEL for every individual that showed an altered pigmentation phenotype?

Subcellular

Figure 5: The authors should state that circular melanosomes are irregular because PMEL helps them elongate. I do see the xanthophore differences, but it would help to add a small indication for a xanthophore with amorphous material and one with concentric lamellae.

Lines 265-267: Authors appear to have repeated above statements.

Figure S6: Include a legend for the micrograph notations. This figure would be improved with comparison to a wild type individual.

Progenitors

It was not clear which morph of corn snake was investigated. I assumed that it was a wild type snake for this section. It would've been nice to see the comparison with single cell RNAseq of a Terrazo embryo, but I understand that the expense of this experiment likely made it not possible. However, it's possible that other genes of importance that may be up- or down-regulated in these chromatophore progenitor cells in Terrazo were missed.

Lines 318-320: It is possible that the samples that the authors took contain a large amount of melanophore line progenitors, as opposed to xanthophore and iridophore lines. See also above comments in the summary on chromatophore progenitors. I would like to see a stronger section in the discussion.

Line 283: (Single-cell RNA sequencing) Incorrect reference to figures - you mean Supplementary Figure S7.

Lines 284, 299, 320, 326: (Single-cell RNA sequencing) Incorrect reference to figures - you mean Figure 6.

Methods

Line 519: (Single-cell RNA sequencing) Incorrect reference to figures - you mean Supplementary Figure S7.

Spatio-temporal Expression

Figure 7: PMEL is present in aggregates at the areas that become blotches in wild type patterns. It develops antero-posteriorly and develops left-right alignment later (~S7).

Comparison with Figure S8 is difficult due to the alternate angle. The authors seem to be detecting PMEL, but not in blotches; however, I find it difficult to say that they are not detecting PMEL at all in Terrazzo when you look at S3-S4. If it is ontogenetic differences, that should be clearer from the authors' writing.

Line 352: (WISH) Incorrect reference to figures. Figure S6 is subcellular, and it does not show anything related to PMEL expression.

Figure 8: The references to the lines are very hard to see, and I initially didn't see what the authors were referring to. It could be that the authors see it better in 3D or real life than in a 2D image, but it's not very convincing.

Discussion

I would like to see a punchier discussion section from the authors. The results are fascinating to pigmentation/color pattern biologists, so I would just like to see a more fleshed out discussion (see: suggestion for summary figure).

Lines 382-383: The authors don't really show that PMEL is required for differentiation of chromatophores - only that it affects expressing the blotching pattern or not (Terrazzo).

Lines 405-408: I agree, but this conclusion does not come through in the Results section (re: WISH).

Future directions should also follow up on ERBB3 and cell counts of the different chromatophores and/or chromatophore progenitors.

Reviewer #3 (Remarks to the Author):

Reviewer #4 (Remarks to the Author):

In the submitted manuscript, the authors attempted to reveal the genetic mechanism of coloration patterns in Terrazzo corn snakes (with striped skin color rather than blotches as wild-type). The authors discovered a causal mutation in PMEL gene whose expression level was significantly downregulated in Terrazzo embryonic tissues. Applying CRISPR-Cas9, they successfully produced PMEL-KO corn snakes which presented the comparable coloration phenotype to Terrazzo and the similarly impacted subcellular structure of melanosomes and xanthosomes, confirming the involvement of PMEL in the patterning process. After that, the authors performed single-cell expression analysis wild-type embryonic dorsal skin, proving PMEL is a marker of chromatophore

progenitors, which can be used to follow the pattern formation during development. At last, results of embryonic WISH suggested the aggregation model of PMEL-expressing cells, that however was absent in Terrazzo embryos, which revealed an important pattern of snake coloration from blotches to stripes. Overall, I believe that the research makes a number of interesting and valuable contributions to reptile coloration that merit publication in Nature Communications, but some issues to be aware of before accepted:

1. Line 125: Have the authors considered the effect of mutations on protein structure?
2. Line 165: Why embryos from the E20 period were selected for this study? And I am also curious as to how the authors were able to isolate the specific dorsal skin tissue during this period? A more detailed description is needed.
3. Line 182: What's the definition of 'significantly downregulated' here (as the very low log₂ fold-change)?
4. Line 185-186: It seems an overstatement to me, especially with low levels of gene expression. The authors need to explain this more and provide more evidence to support this view.
5. Line 261-263: The authors need to do a better job of labelling this change in Fig.5, otherwise it's hard for me to discern the difference (especially in xanthosomes).
6. Line 281: Same query as point 2.
7. Line 284: Fig. 7a should be Fig. 6a. The same applies where it appears later in the section.
8. Could the authors try to restore PMEL expression in KO embryos (like a rescue experiments with injected mRNA) to see if the phenotype is restored, which I think would better help to confirm the function of PMEL.
9. Fig. 4b, c: Note that the figures of lateral are divided by a dotted line, is this an error?
10. Fig.5: It will help to better understand the differences at the cellular level if the comparison of TEM results of the overall chromatophore arrangement can be added (with smaller magnification or different horizons).
11. Line 241-248: I think the results here are interesting, does this suggest that there are other regulatory mechanisms influencing colouration during development and could the authors discuss this briefly?
12. Line263: Given the connections of chromatophore developmental processes, could the authors go further and discuss the possibility of iridophore differences (e.g., arrangement, density, etc.) based on the TEM results of the overall arrangement of chromatophores.
13. Supplementary Figure 5a: In the bottom left corner of the figure there is a WT sample that looks as if it is clustered with Terrazzo, is there any explanation for this?
14. Supplementary Figure 6: Need to add figure notes to help readers better understand cell types.
15. Line 509: Has sequencing been performed on mixed samples of embryonic skin samples (as the amount of embryonic tissue is extremely small)?
16. Line 534: Whether the TEM results are biologically repetitive?
17. Please check the formatting mistake across the whole manuscript. (e.g. Line 293: “)"); species Latin names should be italicized)
18. Supplementary Table 4: The letter 'p' in p-value needs to be italicized, and wherever else it appears across the manuscript.

From blotches to stripes: the role of PMEL in snake coloration pattern
Athanasia C. Tzika*, Asier Ullate-Agote, Pierre-Yves Helleboid and Maya Kummrow

We are very grateful to the four Reviewers for the positive evaluation of our manuscript and for their extensive and constructive comments that allowed us to improve the content and quality of our manuscript. This revision required additional laboratory and bioinformatics analyses.

The most important modifications are listed here in their order of appearance in the text:

1. We provide additional data to explain the clustering of the bulk RNA-seq samples (Supplementary Figure 5).
2. We sequenced the *PMEL* target region from biopsies taken from the *PMEL* mutants and present the electropherograms in the new Supplementary Figure 6.
3. We improved the display and annotation of the TEM images and introduced the new Supplementary Figure 8, where the distribution of chromatophores in the *PMEL* mutant skin is presented.
4. We clarified all mentions of developmental stages.
5. We clarify the sampling methods for the bulk and single-cell transcriptomic experiments.
6. We incorporated the comments of the reviewers on the interpretation of our data.
7. In the manuscript, we refer to our work on leopard gecko chromatophore dynamics and we were previously providing a supplementary figure for the reviewers to compare the data of the two species. This study is now published in PNAS and can be accessed here: <https://www.pnas.org/doi/10.1073/pnas.2400486121>

Point-by-point answers (in blue) can be found below. The modifications in the text are highlighted in yellow. Minor changes/corrections were made through out the manuscript.

Point-by-point Response to Reviewers

Reviewer 1

In this study Tzika et al. identify the gene *PMEL* as corresponding to the domesticated corn snake morph, Terrazzo, by genetic mapping and sequencing, with supporting evidence from CRISPR knockout and gene expression analyses. The authors further demonstrate the predominant expression of *PMEL* in chromatophores by scRNA-seq and they examine roles for *PMEL* in chromatophore development by comparing gene expression in wild-type and Terrazzo/*PMEL* mutants by in situ hybridization and chromatophore ultrastructure by TEM. The study is thorough and the text is clear and I have only minor comments for the authors to consider.

We thank the reviewer for the positive assessment of the manuscript and the constructive comments that we address below and in the revised version of the manuscript.

1. The comparison of *PMEL* F0 knockout phenotypes to Terrazzo and other variants could be revised. Though the authors suggest F0s are closer in phenotype to Terrazzo than other variants, the relative degrees of similarity seem a bit ambiguous, at least to this reviewer. The important point here is that the *PMEL* F0 phenotypes have similarities to that of Terrazzo and so help to establish the correspondence of Terrazzo to *PMEL*. Since mutations in different genes can often give the same phenotypes, anyway, showing the other variants here, and inferring similarities or differences, does not really add new information relevant to the Terrazzo/*PMEL* gene identification per se. In combination with inferences on *PMEL* function (from ISH), however, it is potentially interesting that similarities exist between F0s and other variants, as this would suggest a common mechanism at the cellular level (i.e., a progenitor deficiency in the authors' model). The authors should also clearly state whether the other variants are known to be at loci other than Terrazzo from breeder data (i.e., complementation tests, co-segregation analyses), and if somewhat uncertain one might wonder about sequencing *PMEL* from them to check.

We agree with the reviewer that comparison of the *PMEL* F0 knockout phenotype to other pattern morphs is confusing to the reader given that the genetic characterization of these color morphs has not been published yet.

To answer the reviewer's questions:

- We have identified the genomic interval harboring the Striped causative variant and it is situated on a different scaffold. With additional transcriptomic and functional analyses, we have identified a candidate gene responsible for this morph, which is not *PMEL*. We are currently working on the manuscript detailing these findings.
- Tessera is one of the few dominant color pattern morphs (<https://iansvivarium.com/morphs/#Single%20Dominant>), so complementation crosses are not informative. We are still working on the genetic mapping of this morph.

As all this information is yet unpublished, we decided to remove this part (main text and Figure 4) and focus only on the comparison with the Terrazzo. The comparison with the other morphs will be more appropriate once we have finalized their characterization. Then we will indeed be able to suggest a common mechanism at the cellular level, as the reviewer suggests.

2. The variation in F0 phenotypes is intriguing. Were these zebrafish for example, one would assume that differences among individuals and relative to the original variant reflect underlying mosaicism of the cells involved. Yet the sequencing is said to have revealed homozygosity in most instances with mutation even to the paternal allele post-fertilization. Given the ambiguity I recommend showing some Sanger electropherograms as examples, as many readers will be

used to looking at these in the context of other species for which they might be analyzed computationally to reveal the degree of mosaicism. Additionally, it seems important to assess whether there is indeed any mosaicism in the relevant population of cells, as one could easily imagine that phenotypic heterogeneity reflects clonal expansion of small numbers of escaper cells, which--as the authors carefully point out--might not be evident in shed epidermis. Of course the animals are precious given the difficulty of generating them, and presumed regulatory constraints on using them. But if sequencing of small dermal biopsies were possible it would potentially be informative for interpreting these phenotypes and those of other species in which biallelic knockout has been seen.

We have now included the new Supplementary Figure 6 with electropherograms of the target region for (i) the transfected fibroblasts, (ii) mutated animals with a wild-type phenotype, (iii) mutated animals with stripes, (iv) the unique heterozygous, (v) mutated animals with both blotches and stripes, and (vi) mutated animals with modified blotches. For the later two cases, we extracted DNA from skin biopsies (dermis and epidermis) and sequenced the target region. We observe the same mutations in the anterior and posterior parts of these animals, so they might not be mosaic. As we indicate in the text, their variable phenotype might be due to their genetic background, undetected off-target mutations, or variation in the expression levels of *PMEL* during their development. Obviously, the cells sampled with the biopsies are not the ones that established the pattern during development.

3. At lines 320 and 326 it appears that Figure 6 should be cited; at line 358, suppl figure 8C.

We have verified and corrected all references to figures throughout the text.

4. Imaging of progenitors stained by IHC really would benefit from higher magnification details, sufficient to show individual cells rather than broad (and seemingly diffuse) pattern. Since the authors have these specimens presumably, quantitative assessments of cell numbers in defined regions would seem possible to obtain. These would strengthen the conclusions and would align with the manuscript text that seeks to assess determine the impact "on the number of chromatophores."

Unfortunately, it is not possible to count the cells with the whole mount in situ hybridisations we performed even with greater magnification imaging. Nevertheless, we provide magnifications for the reader to appreciate the differences between wild-type and Terrazzo. As we do not detect any signal with the *PMEL* probe on Terrazzo embryos, we added magnifications from the *MLANA* and *GPNMB* results in Figure 8. In the closeups, we can appreciate that the density of stained cells in Terrazzo is smaller than in the wild-type.

5. At line 281, a brief description of stage S9 would be helpful (e.g. relative to NC migration, pigment cell differentiation). It appears later but the reader could use the information up front in evaluating scRNA-seq results.

As the reviewer suggests, we moved the reference to the supplementary text on staging to this section and added information on the S9 stage.

6. In suppl figure 8, add annotation to indicate Terrazzo for the lower panels.

We modified the new Supplementary Fig. 10 as requested.

7. In figure 4, add to legend a description of the vertical dashed line and image splice/comparisons for b and c lateral views.

An explanation has now been added. The photos on the right side of the vertical dashed lines correspond to the posterior part of the animal, where the pattern is different compared to the anterior part.

8. At line 35, zebrafish might be considered a reference for some fishes, or perhaps ectotherms (a stretch), but not animals; similarly, suggest avoiding at line 374 the term "classical" based on zebrafish markers. As the authors' group recognizes, and is demonstrating nicely, we just don't know enough about other clades or even other fishes. I'd let zebrafish be zebrafish.

We agree with the reviewer, and we modified the text accordingly.

9. Display levels of TEM in figure 5 could reasonably be adjusted to facilitate comparison of panels. The authors might note that adjustments were made for this purpose in the methods, if concerned about seeming to manipulate images.

The display levels of all TEM images were adjusted as requested and a note was added in the methods section.

Reviewer 2

Review: Accepted with minor revisions

A major revision in interpretation is required (noted in the following paragraph). All other suggestions are minor revisions and listed below.

We thank the reviewer for recommending our manuscript with minor revisions. We address each point raised below and in the revised version of the manuscript.

Summary

The authors demonstrate that PMEL is involved in the pigmentation patterning of corn snakes, and changes to PMEL isoforms and/or expression can lead to modified pigment patterns. This result has not been demonstrated before, and it is a noteworthy and interesting result. This work is likely of significance to understanding pigmentation pattern development, as it contributes to our understanding of how PMEL may contribute to pattern development. However, the authors do not address how the work may be of significance to other fields.

Our findings have implications in several fields, such as evolutionary developmental biology, herpetology, and ecology. We now mention this in the Introduction.

Overall, the approach to determining the potential mechanism by which blotches become stripes in corn snakes is very methodical. They take a top-down approach (mapping-by-sequencing) that allows a relatively unbiased identification of candidate genes, and then followed the most likely candidate (PMEL). They tested the phenotypic effect of this candidate using a knockout mutant and other follow-ups to begin understanding the mechanisms underlying these corn snake pigmentation patterns.

Indeed, we have employed several techniques to support our claim that PMEL plays an important role in skin coloration patterning in corn snakes, a non-classical model species.

Their first claim of a disruptive mutation (premature stop codons) in PMEL is present in Terrazo was supported in a single individual, but not necessarily for all Terrazo individuals.

We would like to clarify here that our first claim on the disruptive mutation in PMEL is not based on a single individual. It is based on the genome sequencing of 48 individuals used for the mapping (Supplementary Table 1), and it was further confirmed

in all the individuals used for the isoform Sanger sequencing, the bulk RNAseq and the quantitative PCR.

However, they showed strong and significant down-regulation of that disrupted exon and *PMEL* generally in multiple Terrazo individuals, indicating that *PMEL* is a strong candidate driver of pigmentation patterning in corn snakes despite the earlier low sample size. The authors generated *PMEL* knockouts using CRISPR-Cas that qualitatively have similar subcellular pigmentation structures to Terrazo mutants.

We would like to clarify here that the *PMEL* knockouts we generated share both the subcellular modifications and the phenotype found in Terrazo mutants.

Using single-cell RNA sequencing, the authors conclude that *PMEL* is expressed to varying levels in chromatophore progenitors; furthermore, they state that chromatophore progenitors were reduced in Terrazo individuals. I believe that these claims around the chromatophore progenitors require major revision in interpretation. First, the authors did not perform cell counts on chromatophores or chromatophore progenitors in Terrazo and/or knockout individuals. Their claim of reduced numbers relies on in situ hybridization, but their figures demonstrate changes in ontogenetic *PMEL* expression, not necessarily in chromatophore progenitor numbers.

We would like to clarify here that our hypothesis on the number of chromatophore progenitors is mainly based on our whole-mount in situ hybridisations with the *MLANA* and *GPNMB* probes. We provide greater magnification images in the revised Figure 8 to support further our claim. As we also mention in the relevant comment below, the blue stain seen in Terrazo embryos corresponds to probe trapping, rather than *PMEL* expression (Supplementary Figure 10).

Second, they claim that *PMEL* is present in all chromatophore progenitors, but I believe that they are basing this conclusion on zebrafish chromatophore development. It is important to note that chromatophore differentiation could be different in squamates (or even specifically corn snakes) compared to teleosts.

We would like to clarify here that our claim that *PMEL* is expressed by all chromatophore progenitors is based on our own single-cell data from a corn snake embryo (Figure 6c), rather than findings in the zebrafish. We obviously expect to find differences between the two species, especially given the whole genome duplication in teleosts and the metamorphosis in zebrafish. Note that we observe expression of *PMEL* by all chromatophore progenitors in leopard geckos as well (manuscript now published and accessible here: <https://www.pnas.org/doi/10.1073/pnas.2400486121>).

Furthermore, zebrafish chromatophore differentiation is complex, and melanophore markers decrease in certain chromatophore progenitors ontogenetically as fate specification turns them into xanthophore or iridophore progenitors (see: Subkhankulova et al. (2023) Nature Communications (doi: 10.1038/s41467-023-36876-4)). I find myself thinking that melanophore differentiation is affected (reduced), which has in turn affected the patterning of the other pigment cells. In zebrafish, changes in *Csfl* affected xanthophore differentiation, which led to cascading effects on other chromatophores' differentiation and patterning (Patterson et al. 2014; Nature Communications; DOI: <https://doi.org/10.1038/ncomms6299>). Similarly, changes in *PMEL* could be having cascading effects on differentiation and patterning, but not necessarily on chromatophore numbers, which would require cell counting. Perhaps this section would be improved with a figure in the discussion about *PMEL* expression in chromatophore differentiation and development (in short, a conclusion figure that wraps up their results and/or future testable hypotheses would be great).

Chromatophore differentiation in zebrafish has been extensively studied and continues to be studied by multiple labs around the world, so a large amounts of data already exists. Knowledge on the differentiation of reptilian chromatophores is limited and as the reviewer points out it is probably different from zebrafish. To our knowledge, this manuscript and our recently published article in PNAS on the differentiation of leopard gecko chromatophores are the first attempts to elucidate how chromatophores differentiate in reptiles (<https://www.pnas.org/doi/10.1073/pnas.2400486121>). For this reason, we are not ready to propose a model for the differentiation of corn snake chromatophores. Our continuous work to genetically characterise additional reptilian morphs and our functional analyses by gene-editing will eventually allow us to put forward such a model. Nevertheless, the findings in this manuscript strongly support the importance of *PMEL* in the patterning of corn snake chromatophores and this is the conclusion that we would like to put forward. We include the comment of the reviewer on the possibility of a ‘cascading effect’ in the Discussion.

Finally, their whole in situ hybridization does support *PMEL*-expressing cells gathering in aggregates in wild type but not Terrazo individuals, strongly suggesting that *PMEL* is important for the creation of the blotched patterning in wild type corn snakes.

Indeed, this is the main conclusion of our work.

Suggestions for minor revisions are written below, primarily asking for clarification of methods for replicability and for corrections to figure references in the main text.

Other Suggestions

Introduction

Line 51 include species names where possible

We have now included the species names.

Figure 1: Beautiful photographs. I recommend either labeling dorsal and ventral on the diagram directly or specifying top/bottom in the figure legend.

We modified the figure legend accordingly.

Line 58 A brief description of the mapping-by-sequencing approach (or a reference to a review, such as Candela et al. (2014) *Journal of Integrative Plant Biology*) would be helpful to frame the approach.

Both in the Results and in the Methods sections, we refer to our 2020 PNAS publication where our mapping-by-sequencing protocol is described in detail (Reference 18: Ullate-Agote, A. *et al.* Genome mapping of a *LYST* mutation in corn snakes indicates that vertebrate chromatophore vesicles are lysosome-related organelles. *Proc Natl Acad Sci U S A* (2020) - <https://www.pnas.org/doi/full/10.1073/pnas.2003724117>).

Methods & Results

Description

In both Lines 82-99 of the main text and Lines 17-53 of the supplementary text, it is not clear at what developmental time point body pigmentation appears. Although later, the authors do mention that chromatophore maturation appears to occur after E32 and E40, but I am unclear how pattern development relates to the chromatophore maturation and the timepoints chosen for analyses throughout the article.

As requested by the other reviewers, we have included additional information on the staging of the embryos at the relevant points.

Mapping

Line 102: Could you provide a supplement or reference to previously published work so that others can confirm that it is a single-locus recessive mutation?

We now provide a reference for this claim which was confirmed by the crosses we did for the mapping and transcriptomics analyses, as well as the management of our colony.

For Figures 2 and S2: I'm not sure that I understand how Super-scaffold 85 has more biallelic variants, when it seems like 41126 and 82133 both have a high proportion of biallelic variants co-segregating. I do agree that 85 has the closest to 1, which they state indicates a high proportion are co-segregating. I personally feel that I do not understand Figure 2A and 2B well enough to comment further on this graphical display.

When performing mapping-by-sequencing experiments, we expect high-levels of co-segregation in the genomic interval where the causative mutation resides. This is the case for Super-scaffold 85. Co-segregation is lower for all other scaffolds. Furthermore, we genotyped additional individuals for SNPs on Super-scaffold 85 and we were able to reduce the genomic interval thanks to the recombinant individuals (Supplementary Figure 3).

Lines 159-160: Unclear why they could only sequence the wild type isoform from the individual. Do they mean that it's the only isoform that they could detect at high enough levels for sequencing?

Indeed, other isoforms might be expressed but at levels too low to be sequenced. We clarify this point in the text.

For the Discussion: I would like to see the authors discuss that the premature stop codons occur before the Kringle-like domain and the transmembrane helix, as that would be important for function of PMEL.

We already mention that the M β fragment is modified or absent, but we have now indicated, both in the Results and the Discussion, that it is the Kringle-like domain and the transmembrane helix of the M β fragment which are affected.

Methods

Line 434: What age were the adults and the offspring? Could the offspring be sexed?

In mapping-by-sequencing experiments, the age of the individuals is not relevant, because we sequence genomic DNA. As the Terrazzo mutation is not linked to the sex of the individuals, we did not verify the sex of all the offspring.

Expression

Figure S5A: Based on the PCA for the RNA expression profiles of each sample, one wild type individual seems to closely group with Terrazzo. Is it possible that the individual is heterozygous? I am unsure how homozygous wild type and heterozygotes were distinguished, especially if they have the same pigmentation pattern (a cross, perhaps?). However, I do not see this PCA result affecting the differential gene expression analyses (DGEA), as the DGE profiles cluster by Terrazzo or WT (Figure 3B).

As indicated in the Methods, for the bulk RNAseq analyses we sampled E20 embryos that do not have any pigmentation. We genotyped these embryos by extracting genomic DNA and using the primers already provided in the relevant section (section 'Bulk RNA-seq sampling and analysis') and the Supplementary Table 5.

Line 179-183: Given that ERBB3 affects melanophores, and that the black outlines around the blotches in WT corn snakes are lost in Terrazzo mutants but maintained in some crispants, I'm

surprised that the authors are so quick to dismiss this finding. A change in low expression levels does not mean that the phenotypic effect is also low. For example, Ruzycki et al. (2015; *Genome Biology*; DOI: <https://doi.org/10.1186/s13059-015-0732-z>) found that, although expression correlated with phenotypic severity of their disease model, the effect was based on a threshold of gene expression differences, as opposed to a linear effect – and even a minor difference had an effect. I don't necessarily expect the authors to follow up on this gene for this study, but they should update their paper (add to discussion) to reflect this possibility and consider follow-up studies on this gene.

We agree with the reviewer that the possible involvement of *ERBB3* in the Terrazzo phenotype should not be dismissed, this is why we provide detailed information on its expression levels both from the bulk and the single-cell analyses. As requested, we indicated in the Discussion that the involvement of *ERBB3* in reptilian coloration should be further studied. Note however that *ERBB3* is not expressed in the chromatophore precursors based on our single-cell analyses. We provide the UMAPs in our answer to Reviewer 4 comment 4 (see below).

The authors did not appear to examine gene ontology and pathway expression of the differentially expressed genes (RNASeq). They could perform an enrichment analysis to determine whether the genes are enriched within a similar KEGG pathway, potentially with a package like Enrichr (Chen et al. 2013; *BMC Bioinformatics*; DOI: <https://doi.org/10.1186/1471-2105-14-128>).

We believe that such analyses are not justified, given that very few genes are differentially expressed, and the statistical power of enrichment analyses would be very low.

Methods

Missing information for RNASeq (following MINISEQE guidelines): Was rRNA removed or mRNA selected for (e.g., poly-A tail selection)? What quality checks were performed on the RNA samples (e.g., Nanodrop, Qubit, Bioanalyzer with RIN, FastQC)?

As indicated in the Methods (section 'Bulk RNA-seq sampling and analysis'), we extracted total RNA and then produced a TruSeq Stranded mRNA Library. We have now indicated that all samples had a RIN \geq 9.6. Regarding the processing of the sequencing data, all information is included in the GEO submission.

Knockout

Although the authors did identify that mutating PMEL causes changes to the pigmentation phenotype that resembles Terrazzo, they only looked at crispants and not further generations of knockout mutants. Given that this limitation appears to be due to the long generation time of corn snakes, I do not believe it is necessary for the authors to provide this additional information at this time.

Indeed, we cannot provide information on the offspring of the knockouts before three years.

Methods

I found it unclear how the crispants were genotyped. Did the authors sequence PMEL for every individual that showed an altered pigmentation phenotype?

As indicated in the Methods (section 'Gene-editing with CRISPR-Cas9'), we extracted genomic DNA from the sheds of all the offspring independently of their phenotype and sequenced the target regions of the gRNAs on PMEL (exon 2 for gRNA 204f and exon 3 for gRNA 302r). The gRNAs and the primers are provided in Supplementary Table 5. We provide the electropherograms in the new Supplementary Figure 6.

Subcellular

Figure 5: The authors should state that circular melanosomes are irregular because PMEL helps them elongate. I do see the xanthophore differences, but it would help to add a small indication for a xanthophore with amorphous material and one with concentric lamellae.

We have now annotated this figure and modified the text to clarify the role of PMEL in the elliptical shape of melanosomes.

Lines 265-267: Authors appear to have repeated above statements.

This is not a repetition. We first describe our findings in Terrazzo and then the PMEL knockouts. The results are similar.

Figure S6: Include a legend for the micrograph notations. This figure would be improved with comparison to a wild type individual.

We included a notation in the figure legend (new Supplementary Figure 7). Images of the wild-type individual can be found in Figure 5 and our 2020 PNAS publication (Ullate-Agote & al, 2020 - <https://www.pnas.org/doi/full/10.1073/pnas.2003724117>), as already indicated in the main text.

Progenitors

It was not clear which morph of corn snake was investigated. I assumed that it was a wild type snake for this section.

Indeed, it was a wild-type embryo. We have clarified this in the main text.

It would've been nice to see the comparison with single cell RNAseq of a Terrazzo embryo, but I understand that the expense of this experiment likely made it not possible. However, it's possible that other genes of importance that may be up- or down-regulated in these chromatophore progenitor cells in Terrazzo were missed.

Given that corn snakes are seasonal breeders laying one clutch per year, this experiment is not possible. Furthermore, our bulk RNAseq analyses comparing wild-type and Terrazzo skin (Figure 3) showed that very few genes are significantly down- or upregulated at these stages of development. Among them, we find *PMEL*, *ERBB3*, and *APOH* and we provide all the information on their expression.

Lines 318-320: It is possible that the samples that the authors took contain a large amount of melanophore line progenitors, as opposed to xanthophore and iridophore lines. See also above comments in the summary on chromatophore progenitors. I would like to see a stronger section in the discussion.

We now indicate in the Discussion that progenitors of the other chromatophores might have been missed by our sampling method. We proceed to compare our findings on the corn snakes with the single-cell analyses we performed in leopard geckos (<https://www.pnas.org/doi/10.1073/pnas.2400486121>).

Line 283: (Single-cell RNA sequencing) Incorrect reference to figures - you mean Supplementary Figure S7.

We have verified all figure references in the revised manuscript.

Lines 284, 299, 320, 326: (Single-cell RNA sequencing) Incorrect reference to figures - you mean Figure 6.

We have verified all figure references in the revised manuscript.

Methods

Line 519: (Single-cell RNA sequencing) Incorrect reference to figures - you mean Supplementary Figure S7.

We have verified all figure references in the revised manuscript.

Spatio-temporal Expression

Figure 7: PMEL is present in aggregates at the areas that become blotches in wild type patterns. It develops antero-posteriorly and develops left-right alignment later (~S7). Comparison with Figure S8 is difficult due to the alternate angle. The authors seem to be detecting PMEL, but not in blotches; however, I find it difficult to say that they are not detecting PMEL at all in Terrazzo when you look at S3-S4. If it is ontogenetic differences, that should be clearer from the authors' writing.

In figure S8 (new figure S10), we choose to show a side view of the entire embryos such that the reader can appreciate the absence of staining in the entire length of the body. Regarding S3 and S4 of the Terrazzo embryos, the diffused blue coloration corresponds to probe trapping in the cavities of the embryo, as it can also be seen in the S2 and S3 wild-type embryos in the same figure. The *PMEL* staining corresponds to dots, which presumably correspond to cells, as evident in Figure 7.

Line 352: (WISH) Incorrect reference to figures. Figure S6 is subcellular, and it does not show anything related to PMEL expression.

We have verified all figure references in the revised manuscript.

Figure 8: The references to the lines are very hard to see, and I initially didn't see what the authors were referring to. It could be that the authors see it better in 3D or real life than in a 2D image, but it's not very convincing.

As requested by Reviewer 1, we added close-up photos where the pattern is visible.

Discussion

I would like to see a punchier discussion section from the authors. The results are fascinating to pigmentation/color pattern biologists, so I would just like to see a more fleshed out discussion (see: suggestion for summary figure).

We thank the reviewer for the positive appreciation of our findings. We have updated the Discussion as requested at each comment.

Lines 382-383: The authors don't really show that PMEL is required for differentiation of chromatophores - only that it affects expressing the blotching pattern or not (Terrazzo).

We modified the text accordingly.

Lines 405-408: I agree, but this conclusion does not come through in the Results section (re: WISH).

We have now modified the text in the Results section and we indicate that the patterning process seems to be arrested in the Terrazzo animals.

Future directions should also follow up on *ERBB3* and cell counts of the different chromatophores and/or chromatophore progenitors.

We included a comment on the follow up on *ERBB3* in the Discussion.

Reviewer 3

We thank the reviewer for the assessment of our manuscript and the constructive comments.

Reviewer 4

In the submitted manuscript, the authors attempted to reveal the genetic mechanism of coloration patterns in Terrazzo corn snakes (with striped skin color rather than blotches as wild-type). The authors discovered a causal mutation in *PMEL* gene whose expression level was significantly downregulated in Terrazzo embryonic tissues. Applying CRISPR-Cas9, they successfully produced *PMEL*-KO corn snakes which presented the comparable coloration phenotype to Terrazzo and the similarly impacted subcellular structure of melanosomes and xanthosomes, confirming the involvement of *PMEL* in the patterning process. After that, the authors performed single-cell expression analysis wild-type embryonic dorsal skin, proving *PMEL* is a marker of chromatophore progenitors, which can be used to follow the pattern formation during development. At last, results of embryonic WISH suggested the aggregation model of *PMEL*-expressing cells, that however was absent in Terrazzo embryos, which revealed an important pattern of snake coloration from blotches to stripes. Overall, I believe that the research makes a number of interesting and valuable contributions to reptile coloration that merit publication in Nature Communications, but some issues to be aware of before accepted.

We thank the reviewer for the positive assessment of the manuscript and the constructive comments that we address below and in the revised version of the manuscript.

1. Line 125: Have the authors considered the effect of mutations on protein structure?

Given that the protein structural domains were properly identified by InterProScan, we consider that the tertiary structure of the proteins is unaffected despite the presence of non-synonymous mutations.

2. Line 165: Why embryos from the E20 period were selected for this study? And I am also curious as to how the authors were able to isolate the specific dorsal skin tissue during this period? A more detailed description is needed.

As we have now indicated in the text, E20 is the earliest stage at which the skin can be dissected and early stage chromatoblasts are present. We included a short description of the skin dissection in the methods section.

3. Line 182: What's the definition of 'significantly downregulated' here (as the very low log₂ fold-change)?

As stated in the legends of Figure 3a and Supplementary Figure 5c, *ERBB3* has an adjusted p-value, considering FDR correction, lower than 0.05. We have added this information in the text. The low variance of expression levels between groups makes the log₂ fold-change of -0.49 significant, but *ERBB3* expression levels are very low already in the wild-type (~10 transcripts per million) compared to the main candidate gene *PMEL* at ~140 transcripts per million. It is for this reason, that we consider it as a less likely candidate, although we cannot exclude that it is involved as already stated in the text.

4. Line 185-186: It seems an overstatement to me, especially with low levels of gene expression. The authors need to explain this more and provide more evidence to support this view.

Given the difficulty to obtain a sufficient number of embryos, we performed RNAseq at a single developmental stage. We cannot exclude that *ERBB3* and *APOH* are expressed at higher levels at another developmental stage (before or after) and have some impact on the phenotype. We now include in Supplementary Figure 5, the Counts per Million calculation of these two genes that we calculated from the single-cell data for the wild-type. For the single-cell experiment, we sampled at a later stage. *ERBB3* and *APOH* expression remains low at this stage as well. We also provide here for the reviewer the UMAPs for these two genes, which are barely expressed by chromatophores.

In conclusion, we don't have any evidence to support the possibility that *ERBB3* and *APOH* are involved in the Terrazzo phenotype, but it is a hypothesis that we cannot reject. We modified the text accordingly.

5. Line 261-263: The authors need to do a better job of labelling this change in Fig.5, otherwise it's hard for me to discern the difference (especially in xanthosomes).

As recommended by Reviewer 1, we adjusted the display levels of this figure to facilitate comparisons. We also added annotations for the modifications in melanophores and xanthophores.

6. Line 281: Same query as point 2.

The text was modified following also the recommendation of Reviewer 1.

7. Line 284: Fig. 7a should be Fig. 6a. The same applies where it appears later in the section.
We have verified and corrected all references to figures throughout the text.
8. Could the authors try to restore *PMEL* expression in KO embryos (like a rescue experiments with injected mRNA) to see if the phenotype is restored, which I think would better help to confirm the function of *PMEL*.
Unfortunately, this type of experiments is not possible for the following reasons:
- (i) We currently only have access to juvenile *PMEL* KO. These animals will reach sexual maturity in 2027. It is only then that we will be able to breed them and obtain *PMEL* KO embryos.
 - (ii) The corn snake eggs have a soft shell making the manipulation of the embryo in ovo, for example by windowing the egg, impossible.
 - (iii) Ex ovo development of reptilian embryos has been set up only for a couple of species because it is difficult to maintain the embryos alive for long periods (60 days of incubation for the corn snake).
9. Fig. 4b, c: Note that the figures of lateral are divided by a dotted line, is this an error?
This was an omission from our side. As also requested by Reviewer 1, we have now added an explanation in the figure legend.
10. Fig.5: It will help to better understand the differences at the cellular level if the comparison of TEM results of the overall chromatophore arrangement can be added (with smaller magnification or different horizons).
In Supplementary Figure 6, we already provide TEM images at a smaller magnification. These images represent different regions/colors of skin from a Terrazzo individual. We have now added the new Supplementary Figure 8, with images from a *PMEL* knockout.
11. Line 241-248: I think the results here are interesting, does this suggest that there are other regulatory mechanisms influencing colouration during development and could the authors discuss this briefly?
Indeed, we expect different mechanisms to be involved in the differentiation and the patterning of the chromatophores. We have modified the text accordingly.
12. Line263: Given the connections of chromatophore developmental processes, could the authors go further and discuss the possibility of iridophore differences (e.g., arrangement, density, etc.) based on the TEM results of the overall arrangement of chromatophores.
From the overall arrangement of the chromatophores, we assume that the Terrazzo stripes correspond to the wild-type blotches. We have not observed anything particular for the iridophores, keeping in mind that the TEM images correspond to small skin sections. We have included these statements in the text.
13. Supplementary Figure 5a: In the bottom left corner of the figure there is a WT sample that looks as if it is clustered with Terrazzo, is there any explanation for this?
We provide additional information in Supplementary Figure 5, which shows that the PCA clustering is driven by the sex of the embryos, rather than their coloration which is not sufficiently developed at this stage.
14. Supplementary Figure 6: Need to add figure notes to help readers better understand cell types.
We have now included these notes.

15. Line 509: Has sequencing been performed on mixed samples of embryonic skin samples (as the amount of embryonic tissue is extremely small)?

We obtained sufficient material from a single sample, so it was not necessary to pool. We indicated this in the text.

16. Line 534: Whether the TEM results are biologically repetitive?

Based on the similarities we observe between the one-year old and the adult Terrazzo individuals, the TEM results are reproducible. We have observed reproducibility of the TEM imaging with samples from wild-type and other morphs, for example presented in Ullate-Agote & al, 2020 (<https://www.pnas.org/doi/full/10.1073/pnas.2003724117>).

17. Please check the formatting mistake across the whole manuscript. (e.g. Line 293: “)"); species Latin names should be italicized)

We have introduced these corrections.

18. Supplementary Table 4: The letter ‘p’ in p-value needs to be italicized, and wherever else it appears across the manuscript.

We have introduced this correction.

REVIEWERS' COMMENTS

Reviewer #1 (Remarks to the Author):

The authors have thoughtfully and productively addressed my comments and suggestions.

Reviewer #2 (Remarks to the Author):

The authors have appropriately addressed the Reviewer's concerns. Huge congrats on a very cool contribution.

Reviewer #3 (Remarks to the Author):

Reviewer #4 (Remarks to the Author):

The authors have done an exceptional job on responding and have addressed all my concerns.